# Size and Shape Optimization of a Guyed Mast Structure under Wind, Ice and Seismic Loading

**Raffaele Cucuzza** [1] [iD]**, Marco Martino Rosso** [1] [iD]**, Angelo Aloisio** [2,*] [iD]**, Jonathan Melchiorre** [1] [iD]**, Mario Lo Giudice** [1] **and Giuseppe Carlo Marano** [1] [iD]

1 Department of Structural, Geotechnical and Building Engineering, Politecnico di Torino, Corso Duca Degli Abruzzi, 24, 10128 Turin, Italy; raffaele.cucuzza@polito.it (R.C.); marco.rosso@polito.it (M.M.R.); jonathan.melchiorre@polito.it (J.M.); logiudice.mario@polito.it (M.L.G.); giuseppe.marano@polito.it (G.C.M.)

2 Civil Environmental and Architectural Engineering Department, Università Degli Studi dell'Aquila, via Giovanni Gronchi n.18, 67100 L'Aquila, Italy

* Correspondence: angelo.aloisio1@univaq.it

**Abstract:** This paper discusses the size and shape optimization of a guyed radio mast for radio-communications. The considered structure represents a widely industrial solution due to the recent spread of 5G and 6G mobile networks. The guyed radio mast was modeled with the finite element software SAP2000 and optimized through a genetic optimization algorithm (GA). The optimization exploits the open application programming interfaces (OAPI) SAP2000-Matlab. Static and dynamic analyses were carried out to provide realistic design scenarios of the mast structure. The authors considered the action of wind, ice, and seismic loads as variable loads. A parametric study on the most critical design variables includes several optimization scenarios to minimize the structure's total self-weight by varying the most relevant parameters selected by a preliminary sensitivity analysis. In conclusion, final design considerations are discussed by highlighting the best optimization scenario in terms of the objective function and the number of parameters involved in the analysis.

**Keywords:** guyed mast; structural optimization; genetic algorithm; structural design

## 1. Introduction

Guyed masts are extensively used in the telecommunications industry, and the size, shape, and topology optimization can significantly benefit their transportation and installation. The main loads acting on guyed mast structures arise from wind [1,2], earthquakes [3–6], sudden rupture of guys [7], galloping of guys [8], and sudden ice shedding from ice-laden guy wires [9].

Their optimization must fulfil several requirements under ultimate and service limit states [10]. Specifically, service limit states are crucial for guyed mast structures due to high-amplitude oscillations caused by their high deformability. In some cases, these vibrations have led to a signal loss caused by excessive displacement and rotation of the antennas and, in other cases, have resulted in permanent deformation or failure. Therefore, size optimization of the guyed mast structure represents a challenging task since the increment of the performance ratio of the materials should be counterbalanced by an adequate lateral stiffness to reduce high-vibration drawbacks [11].

Saxena [12] reported several happenings where heavy icing combined with moderate wind resulted in severe misalignment of towers and complete failure. Novak et al. [13] showed that ice accumulation on some parts of the guy wires and moderate winds could lead to the guy galloping, resulting in unacceptable stress levels throughout the structure. The main topics investigated in the field of guyed structures can be summarized as follows:

- Structural design. Several researchers investigated the dynamic response of guyed mast structures through experimental tests and numerical modeling to derive design

approaches and recommendations [14–16]. In particular, there are studies dealing with the dynamic identification and accurate estimate of the wind loads [17–21].
- Nonlinear dynamics. The proneness to global and local instabilities challenged several scholars to estimate and predict the nonlinear behaviour of guyed masts [22–26].
- Structural optimization. The need for guyed structures that are easy to install and transport challenged several scholars to optimize their shape in order to reduce the structural mass without reducing the lateral stiffness and prevent instability phenomena [27].
- Structural control. There are some attempts of control methods to reduce vibrations in mast-like structures [28–30]. Among others, Blachowski [31] proposed the use of a hydraulic actuator to control cable forces in guyed masts using Kalman filtering.

This paper tackles the size and shape optimization of guyed mast structures. A video of the considered structure is available in Supplementary Material. Since the first attempts by Bell and Brown [32], many engineers attempted to optimize guyed masts under wind loads using deterministic global optimization algorithms. However, as pointed out by [27], this approach leads to local optimum points, since each design variable was considered separately. Thornton et al. [33] and Uys et al. [34] proposed general procedures for optimizing steel towers under dynamic loads. To the author's knowledge, Venanzi and Materazzi [35] were the first to implement a multi-objective optimization method for guyed mast structures under wind loads using the stochastic simulated annealing algorithm for size optimization. The objective function implemented by [35] included the sum of the squares of the nodal displacements and the in-plan width of the structure. Zhang and Li [36] attempted to achieve both shape and size optimization in a two-step procedure using the ant colony algorithm (ACA). Cucuzza et al. [37] proposed an alternative approach in which the multi-objective optimization problem has been reduced to a single-objective problem through suitable parameters. Luh and Lin [38] were challenged in achieving the topology, size, and shape optimization of guyed masts using a modification of the binary particle swarm optimization (PSO) and the attractive and repulsive particle swarm optimization.

This paper discusses the size optimization of guyed masts using a genetic algorithm (GA) by considering different design scenarios (e.g., Cucuzza et al. [37] and Manuello et al. [39]). Kaveh and Talatahari [40] noticed that the particle swarm optimization (PSO) is more effective than ACA and the harmony search scheme for optimizing truss structures. However, Deng et al. [41] and Guo and Li [42] were successful in optimizing tapered masts and transmission towers using modifications of genetic algorithms (GA). Moreover, Belevivcius et al. [27] attempted the topology-sizing optimization problem of the guyed mast as a single-level single-objective global optimization problem using GAs.

Therefore, given the numerous successful solutions of guyed masts using GAs, the authors chose to investigate the size optimization of a guyed mast structure using GAs. Following [35], this paper focuses on the size optimization by considering eight possible design scenarios. The purpose of the present paper is two-fold. Firstly, this work aims at achieving a size optimization on a real application case adopting structural verification according to Eurocode 3. During the load evaluation phase, detailed analyses have been conducted, including wind, ice, and seismic actions and the verifications against instabilities. Secondly, the computational intelligence procedure adopted by the authors allowed the investigation of several scenarios simultaneously. As a result, the parameters that mainly affected the design process have been detected to provide preliminary indications to engineers in the practical design of similar structural typologies. Furthermore, the considered case study may represent a benchmark case for validating the reliability and accuracy of alternative numerical approaches. Therefore, the paper is organized as follows. After the case study description and the FE model, the authors introduce the first numerical results and the outcomes of the size and shape optimization.

## 2. Case Study

The considered structure is a guyed radio mast. It is a thin, slender, vertical structure sustained by tension cables fixed to the ground and typically arranged at 120° between each other.

The main body is a single central column made of tube profiles or truss systems when a high elevation must be reached, see Figure 1. More than one set of cables is placed at different elevations to prevent instability phenomena. Guyed towers are usually built for meteorological purposes or to support radio antennas, such as the one considered in this research. In particular, this structure can be used for a limited time during an event or maintenance of primary transmission towers. Therefore, it is also called a temporary base transceiver station (BTS), typically adopted to supply the immediate service. Sporting events, concerts, motor racing, military camps, and emergency events are typical examples of temporary BTS applications. The BTS is usually mounted on a moveable platform called the shelter.

The considered structure is located in Bassano Del Grappa, in the north of Italy, at a 129 m elevation from the sea level. The surrounding area is low-urbanized, with no relevant obstacles to the wind loads. The total height of the mast is 30.00 m. It is sustained by a central pole where 21 cables are fixed, see Figure 2. Other structural elements with rectangular cross-sections are used to create truss systems connecting cables and the central pole.

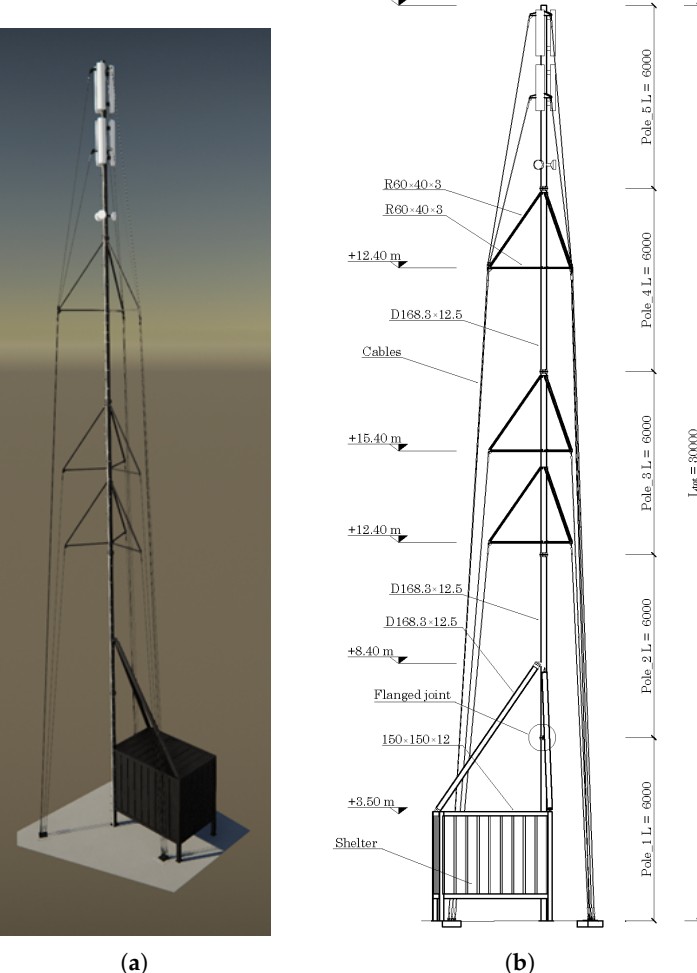

(a)          (b)

**Figure 1.** (**a**) Render model realized using Tekla Structures. (**b**) Technical drawing of the structure investigated with dimensions in mm.

The central pole consists of five circular hollow steel profiles with flanged joints and 6 m in length. All connections are bolted, as well as those connecting the cables to the pole. The shelter is a steel box devoted to partially sustaining the structure and hosting electronic equipment. It is usually mounted on a moveable platform.

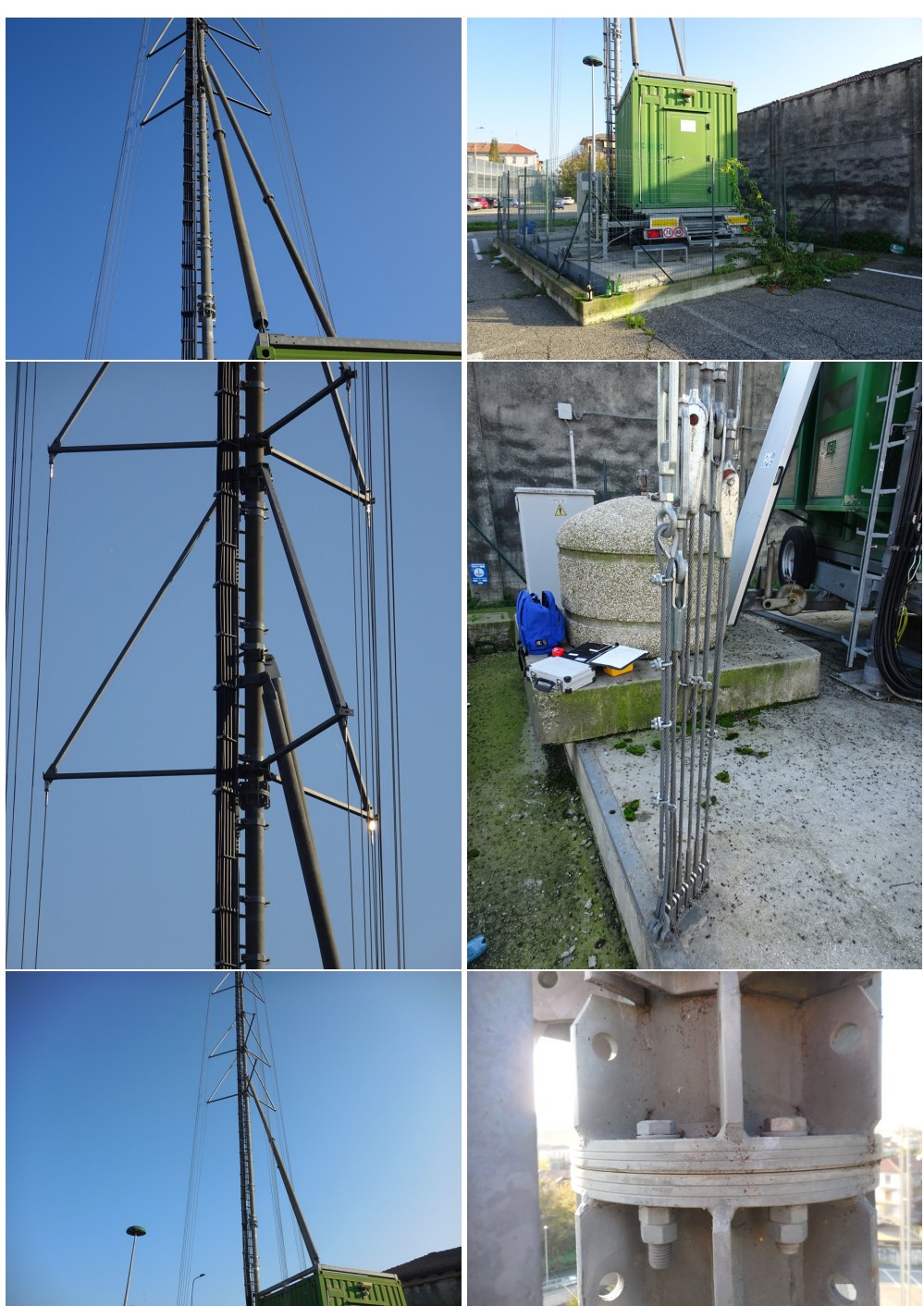

**Figure 2.** Pictures and details of the considered structure.

## 3. Load Analysis

This section details the loads acting on the structures, from the dead to the variable loads. According to the Italian Standard Regulation NTC2018, the load combinations of the actions have been evaluated at the ultimate limit state (ULS) and, for seismic conditions, at the life safety (LS) limit state. In Appendix A, Table A4 illustrates the most critical combinations for both static and dynamic configurations. Partial safety factors $\gamma$ and

combination coefficients $\psi$ were adopted in order to consider maximization (positive sign) or minimization (negative sign) of effects both for vertical and horizontal actions.

### 3.1. Dead Loads

The structure is made of steel S355, whose mechanical stress-strain behaviour is depicted in Figure 3, and the characteristics are listed here: $f_{us}$ = 510 MPa, $f_{ys}$ = 355 MPa, $E_s$ = 210,000 MPa, which are the ultimate and yielding stresses and Young's modulus, respectively.

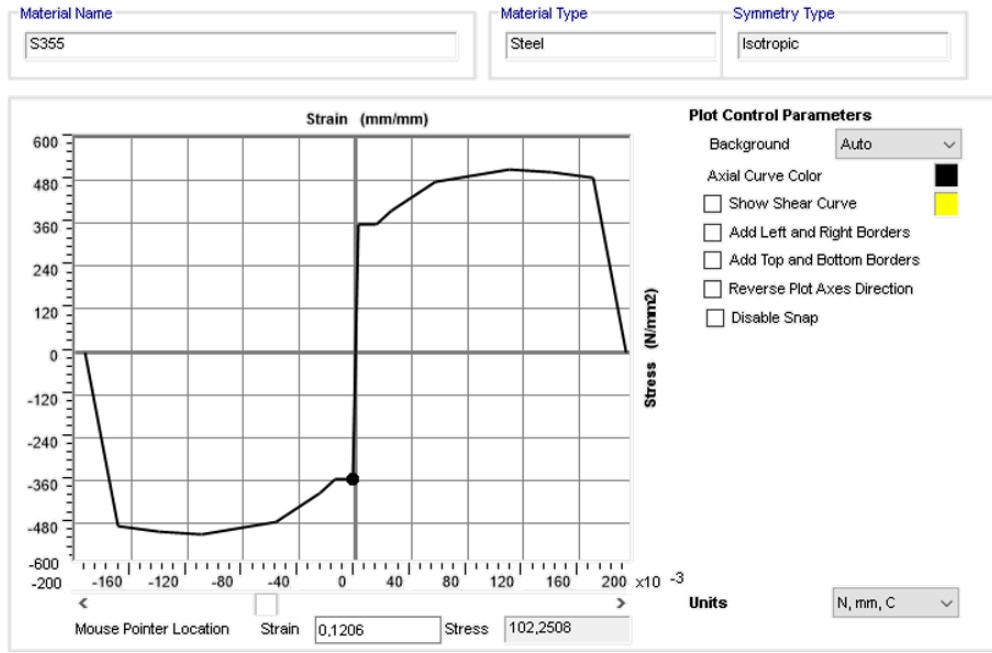

**Figure 3.** Mechanical stress–strain behaviour of steel S355 implemented in SAP2000.

The cables are made of galvanized steel consisting of 6 strands (216 wires) with an independent metal core (49 wires). The main characteristics are illustrated in Table 1.

**Table 1.** Technical specifications of the steel ropes.

| **Steel Ropes (Cables)** | | | | |
|---|---|---|---|---|
| Model | | | $6 \times 36WS + IWRC/265$ wires | |
| Construction pattern | | | $6 \times (14 + (7 + 7) + 7 + 1) + (7 \times 7)$ | |
| Winding direction | | | right cross | |
| Material | | | galvanized steel | |
| Resistance | | | 1170 N/mm$^2$–180 kg/mm$^2$ | |
| Cable diameter | Weight | Area | Wire diameter | Load to failure |
| [mm] | [kg/m] | [mm$^2$] | [mm] | [kN] |
| 16 | 1.36 | 173.25 | 0.91 | 161 |
| 18 | 1.67 | 212.74 | 1.03 | 204 |
| 20 | 2.02 | 257.32 | 1.14 | 252 |
| 22 | 2.41 | 307.01 | 1.26 | 305 |

The structure investigated consists of a few types of elements, as indicated in Table 2. Dead loads are calculated from the weight per unit length of each member.

**Table 2.** Computation of the dead loads.

| Computation of Dead Loads | | | | | |
|---|---|---|---|---|---|
| Profile [mm] | | w [kg/m] | Length [m] | n° | Wtot [kg] |
| Circular | D168.3 × 12.5 | 48 | 6 | 5 | 1440 |
| | D168.3 × 12.5 | 48 | 5.65 | 2 | 543 |
| Rectangular | 60 × 40 × 3 | 4.35 | 3.16 | 9 | 124 |
| | 60 × 40 × 3 | 4.35 | 1.8 | 9 | 71 |
| | 100 × 40 × 3 | 6.13 | 0.45 | 6 | 17 |
| Rope | D16 | 1.3667 | 12.45 | 3 | 51 |
| | D16 | 1.3667 | 15.44 | 3 | 63 |
| | D16 | 1.3667 | 24.43 | 9 | 300 |
| | D16 | 1.3667 | 5.76 | 3 | 24 |
| | D16 | 1.3667 | 8.46 | 3 | 35 |
| | | | | 2651 Kg | |

The non-structural dead loads originate from the wiring weight and the steel ladder for inspection and maintenance. This load results in 0.3 kN/m. Antennas and parabolas represent the weight of the equipment. Two groups of three antennas are located at 26.00 and 29.25 m in height, with a 120° in mutual spacing. The first one is the model AOC4518R7v06 produced by Huawei®. The second one is the model 6888670N manufactured by Amphenol®. Finally, there are three parabolas located at 23.15 m height, spaced 120° apart from each other, 30 cm in diameter. Tables 3 and 4 detail the weight of the equipment and the non-structural dead loads.

**Table 3.** Weight of equipment, H, W, and D stand for height, width, and depth.

| Typology | Model | No | Elevation [m] | H×W×D [mm] | Self-Weight [kg] | Clamps [kg] | Total [kg] |
|---|---|---|---|---|---|---|---|
| Antenna | AOC4518R7v06 | 3 | 29.25 | 1509 × 469 × 206 | 39.3 | 2 × 5.8 | 153 |
| Antenna | 6888670N | 3 | 26 | 1997 × 305 × 163 | 32 | 2 × 3.9 | 119 |
| Parabola | n.d | 3 | 23.15 | Diameter = 300 | 15 | 2.2 | 51.6 |

**Table 4.** Non-structural dead loads.

| Item | qk [kN/m] | Qk [kN] |
|---|---|---|
| Steel ladder, other | 0.3 | - |
| Antenna | - | 1.53 |
| Antenna | - | 1.19 |
| Parabolas | - | 0.52 |

### 3.2. Variable Loads

In this section, the detailed load modeling phase, for each variable load considered, is described. With specific reference to the wind action evaluation, the drag and lift forces are calculated according to the CNR-DT 207 R1/2018 [43]. The relationship between inertia and viscous forces, i.e., how wind load impacts to the surface, is taken into account with the Reynold's number $R_e$ with the following expression:

$$R_e(z) = \frac{l \cdot v_m(z)}{\nu} \tag{1}$$

where $z$ is the elevation, $l$ is the characteristic length, $v_m$ is the averaged wind speed, while $\nu$ is the kinematic viscosity of air ($\nu = 15 \times 10^{-6}$ m²/s).

### 3.2.1. Maintenance and Repairing Loads

Following the Italian national recommendations [44], it is supposed that a typical situation of inspection or maintenance is performed by an operator working on the steel ladder. A concentrated load of 120 kg is applied at the top of the tower. Despite that, it is reasonable to believe that the operator could work by using a basket elevator, without loading the structure.

### 3.2.2. Wind Loads

The wind action was evaluated according to the Italian recommendations in [43]. Firstly, the peak kinetic pressure ($q_p$) was evaluated as follows:

$$q_p = \frac{1}{2} \cdot \rho \cdot v_r^2 \cdot c_e(z) \tag{2}$$

where $p$ is the kinetic pressure, while:

- $\rho$ is the air density;
- $v_r^2$ is the reference wind velocity;
- $c_e$ is the exposure coefficient, varying with the elevation $z$ of the structure.

For this purpose, the equivalent longitudinal or drag forces, $f_D$, and transverse or lift force, $f_L$, are evaluated as follows:

$$f_{drag} = q_p(z) \cdot l \cdot c_{drag}; \qquad f_{lift} = q_p(z) \cdot b \cdot c_{lift} \tag{3}$$

where

- $q_p(z)$ is the *peak kinetic pressure* evaluated at height $z$;
- $l$ is the characteristic element size;
- $b$ is the reference transverse dimension of the section;
- $c_{drag}$ and $c_{lift}$ are the longitudinal and transverse dynamic coefficients.

Drag D and Lift L forces are reported in Tables A2 and A3.

### 3.2.3. Ice Load

Ice and snow attached to the structural surface can significantly increase the variable loads in flexible and light structures. In particular, the radio mast is very sensitive to changes in the wind-exposed surface. In addition, the ice covering can increase the volume and the surface of the structural elements more than twice due to the low thickness of the central pole. The recommendations in [43] provide several scenarios for ice coverings. In the absence of more detailed evaluations, it is customary to consider an ice sleeve formation that is 12.5 mm thick. After the estimate of the wind loads, the influence of the ice sleeve formation on the structure is considered by assuming an additional exposed surface equal to 15% of the original one.

### 3.2.4. Seismic Action

Seismic action is evaluated according to the Italian seismic hazard map [44]. A linear dynamic analysis with seismic elastic response spectrum corresponding to the service limit state was carried out. Specifically, seismic actions are considered as acting independently in the X and Y plane directions.

The elastic response spectrum considered in the analysis was calculated by considering the topographic category of the site and geometry of the building (Figure 4). The first 33 vibration modes of the structure are included in the analysis, to reach 85% of the total participating mass according to the national regulations in [44]. The mass participating ratios are listed in Appendix A.

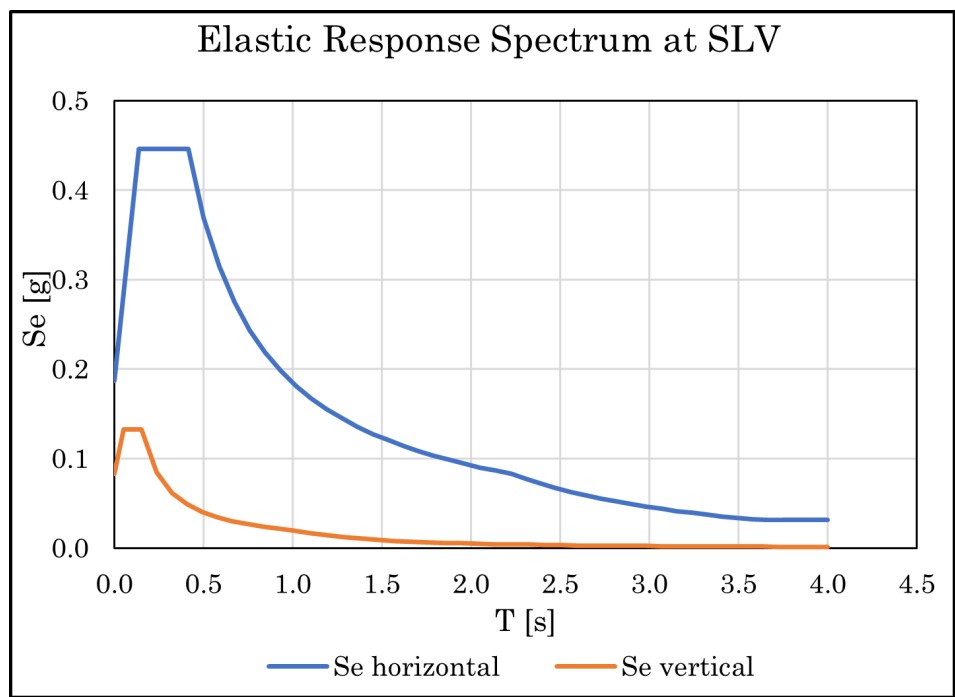

**Figure 4.** Elastic response spectrum corresponding to the service limit state (SLV), where *Sa* is the spectral acceleration.

## 4. Finite Element Modeling

The structural model was developed using two different element types: beams and cables. Beam elements model the main pole and all structural elements except for the cables. They possess the geometric and material properties of structural elements. The beam elements are used to model the main pole and secondary elements. Moreover, except for the main pole, rotation releases are applied at the ends in order to consider no flexural rigidity, as occurring for trussed structures.

Cable elements are used to simulate the steel ropes. Cable elements undergo large displacements that give rise to geometric nonlinearities. Therefore, the equilibrium of the cables is considered in the deformed configuration using SAP2000. As a result, the structural behaviour of guyed towers can be highly nonlinear, especially for low pre-tension cables, which are prone to large displacements. On the contrary, the nonlinear behaviour becomes less pronounced by increasing the pre-tension, resulting in high compression levels and minor instability effects. This paper considers the envelope of the maximum and minimum responses associated with each load condition.

Figure 5 plots the three modes with a higher mass participation ratio. These are the 10th, 11th, and 12th modes obtained from the dynamic analysis of the mast structure with the dead loads. On the contrary, the first modes arising from the dynamic analysis have lower mass participation factors and are characterized by local deformation of the structural elements. The 10th, 11th, and 12th modes are the first modes exhibiting the global deformation of the mast structure.

X and Y identify the in-plane orthogonal directions. The 10th and 11th modes have an approximate 26% mass participation factor in the Y and X directions, respectively. The natural period is very low and at approximately 0.4 s. The 13th mode is mainly torsional with nearly a 7 and 4% mass participation factor in the X and Y directions.

Figure 6 shows the positive (in dark green and purple) and negative (in red and light green) maximum and minimum envelopes of the axial, shear forces, and bending moments acting on the structural elements. Figure 7 plots the performance ratios of all structural elements except for the cables. The performance ratio is the ratio between the maximum stress in the structural element and the yielding stress. The performance ratios are defined by the colour map next to Figure 7. The plots highlight the presence of a structural element

in the first half of the central pole with a high-performance ratio, depicted in yellow. The first section of the central pole has a low performance ratio, lower than 0.25. After the section with a performance ratio in the range 0.4–0.65, the following sections fall in the range 0.25–0.4 and are coloured in green. The top sections of the central pole are not significantly stressed, with a performance ratio of 0–0.25. The bracings have low stress, plotted in cyan, with performance ratios of 0–0.25.

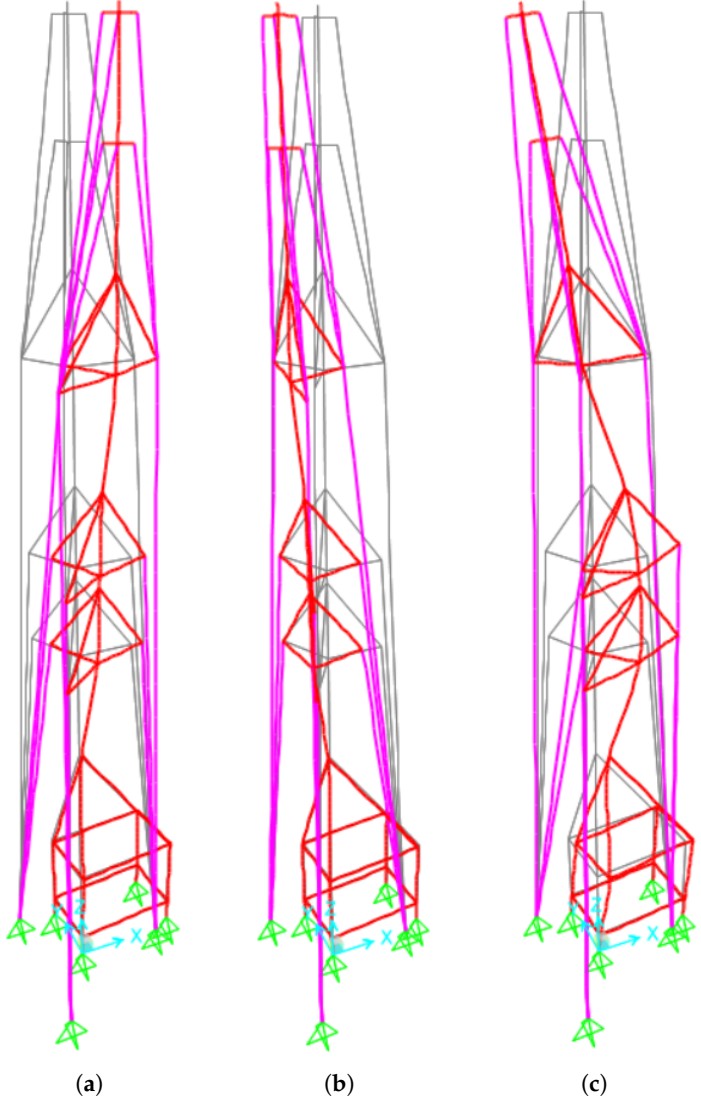

**Figure 5.** (**a**) Mode 10th—Ts = 0.437 s—mass participation ratio X = 9.6%, Y = 26.2%; (**b**) Mode 11th—Ts = 0.434 s—mass participation ratio X = 26.4% Y = 9.2%; (**c**) Mode 12—Ts = 0.206 s—mass participation ratio X = 7.2% Y = 4.4%.

Figure 8 shows the maximum displacements in the X ($u_1$), Y ($u_2$) directions and their combination ($u_t$) at the service limit state. The maximum displacement is located at the top of the tower, in particular at joint 6 (z = 30.00 m), with a maximum displacement equal to $u_t$ = 18.7 mm.

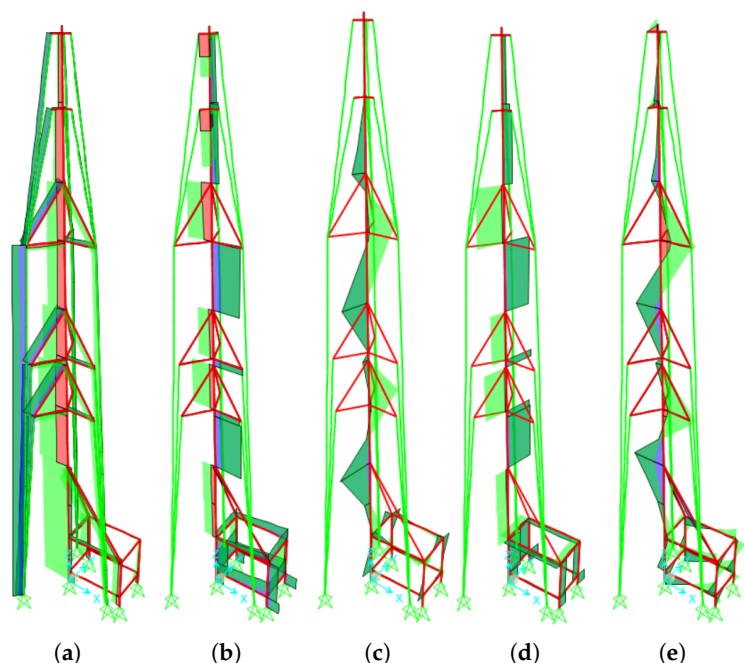

**Figure 6.** (**a**) Axial force, (**b**) shear force ($V_2$), (**c**) bending moment ($M_2$), (**d**) shear force ($V_3$), (**e**) bending moment ($M_3$).

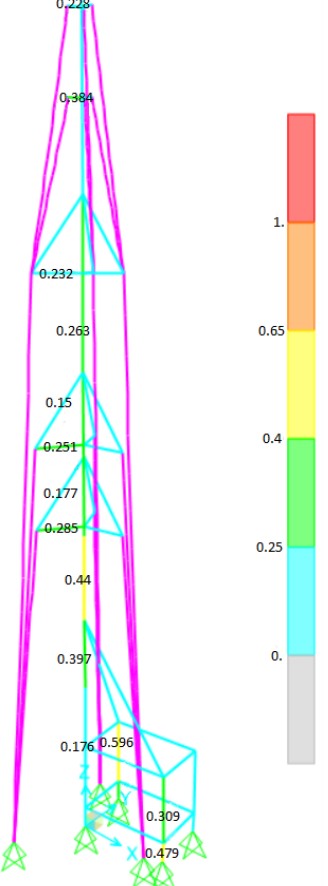

**Figure 7.** Performance ratios of the pole before optimization. Cables are depicted with magenta colour because their performance ratios are not included in the current representation.

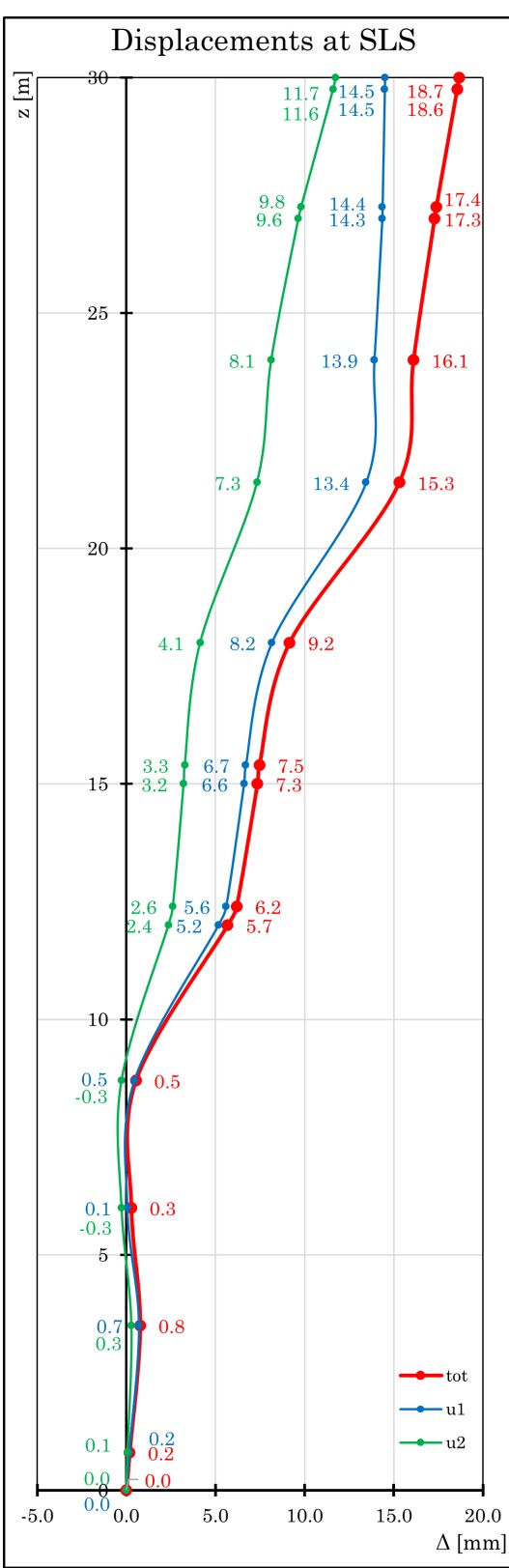

**Figure 8.** Displacements vs. elevation at the service limit state in two in-plane orthogonal directions ($u_1$, $u_2$) and their combination (tot).

## 5. Structural Optimization

In optimization problems, the main goal is to find the best conditions in terms of the optimal set of design parameters collected in the design vector $x$, which minimizes an objective function (OF) $f(x)$ [45–47]. These problems can be categorized into single-objective or multi-objective based on the number of OFs involved, and a further classification is based on the presence (or not) of constraints [48–50]. In the structural optimization field, it is common to deal with constrained optimization, whose general statement is [51]:

$$
\begin{aligned}
&\min_{x \in \Omega}\{f(x)\} \\
\text{s.t.} \quad &g_q(x) \leq 0 \quad \forall q = 1, \ldots, n_q \\
&h_r(x) = 0 \quad \forall r = 1, \ldots, n_r
\end{aligned}
\tag{4}
$$

where $x = \{x_1, \ldots, x_j, \ldots, x_n\}^T$ is the design vector to be optimized, whose terms are limited into a hyper-rectangular multidimensional box-type search space domain of interest denoted as $\Omega$, given by the Cartesian product of the range of interest of each $j$-th of each design variable bounded in $[x_j^l, x_j^u]$, $\Omega = [x_1^l, x_1^u] \times \ldots \times [x_j^l, x_j^u] \times \ldots \times [x_n^l, x_n^u]$. The term $g_q(x)$ in (4) denotes inequality constraints whereas $h_r(x)$ are equality ones, which further reduce the feasible search space inside $\Omega$. In structural optimization, it is typical to deal with inequality constraints, and a common goal is to minimize the global cost of the structure. Since this involves many terms, the main attempt is minimizing the self-weight of the structure, indirectly connected to material cost, i.e., material usage and natural resources consumption [51]. Several strategies have been developed over the years to handle constraints [52–54]. In the present work, the penalty function-based approach was implemented due to its simplicity, allowing converting the problem with OF $f(x)$ into a new unconstrained version $\phi(x)$:

$$
\min_{x \in \Omega}\{\phi(x))\} = \min_{x \in \Omega}\{f(x) + H(x)\}
\tag{5}
$$

where $H(x)$ is the penalty function. Adopting a static-penalty strategy, $H(x)$, assume this form [55,56]

$$
H_s(x) = w_1 H_{NVC}(x) + w_2 H_{SVC}(x)
\tag{6}
$$

where $H_{NVC}$ is the number of violated constraints and $H_{SVC}$ is the sum of all violations:

$$
H_{SVC}(x) = \sum_{p=1}^{n_p} \max\{0, g_p(x)\}
\tag{7}
$$

$w_1$ and $w_2$ are the violation control parameters, whose numerical values are assumed equal to $w_1 = w_2 = 100$ following [55].

In the current study, the authors carried out a parametric study on the design variables of the guyed mast. This fact has led to eight different scenarios, summarized in Table 5. In addition, the starting initial values of the design parameter are listed in Table 6, while the general optimization workflow is illustrated in Figure 9. To compare the results, the focus is related only to the performance ratios PR of the central pole of the guyed radio mast, being the pole the most stressed element. It consists of five segments 6.00 m long with the same cross-section. Thus, starting from the ground level:

1. Pole$_1$ (0.00 to 6.00 m);
2. Pole$_2$ (6.00 to 12.00 m);
3. Pole$_3$ (12.00 to 18.00 m);
4. Pole$_4$ (18.00 to 24.00 m);
5. Pole$_5$ (24.00 to 30.00 m).

Starting with a constant diameter of the cross-section for the pole, at the end of the optimization, it is advisable to find a tapered solution following a linear relationship with

the height, as represented in Figure 10f. Accordingly, it is possible to shape the pole cross-section with two design variables described by the bottom $\Phi_i$ and top $\Phi_f$ diameters. In the following, the different scenarios obtained from the parametric study based on the design variables involved in the optimization problem are described:

- Scenario A: this scenario involves the diameter $\Phi$, as a sole variable, in the attempt to reduce the material consumption with a constant pole cross-section diameter with the height, as illustrated in Figure 10a.
- Scenario B: this scenario attempts to refine the previous case by adopting a tapered solution for the pole, by using the bottom $\Phi_i$ and the top $\Phi_f$ diameters, as represented in Figure 10b.
- Scenario C: further improvements are considered concerning scenario B by adding the cable pre-stressing force $F$ as a variable of the optimization, as represented in Figure 10c.
- Scenario D: further improvements are considered to scenario B by using a unique value for the pole thickness $t$ of the tapered elements of the pole, as represented in Figure 10c.
- Scenario E: further improvements are considered with respect to scenario B by optimizing both cable pre-stressing force $F$ with a unique value of thickness $t$ for the tapered elements of the pole, as represented in Figure 10e.
- Scenario F: from the structural analysis of scenario E, it is possible to point out how the linear law for the tapering forces to use a larger section where it is not necessary. Elements 2 and 3 are the most stressed ones. Therefore it is possible to further refine scenario E by considering a thickness value for the intermediate pole elements $t_{inter}$ and a different thickness for the other extremal pole elements $t_{e}nds$.
- Scenario G: in this scenario, the five different thickness values only have been governed for every pole element $\{t_1, t_2, t_3, t_4, t_5\}$ for a constant diameter solution with height, as depicted in Figure 10f.
- Scenario H: in this last scenario, a complete approach involves both the tapered solution by governing the initial bottom $\Phi_i$ and the final top $\Phi_f$ diameters, the five values of thickness for every pole element $\{t_1, t_2, t_3, t_4, t_5\}$, and even the cable pre-stressing force.

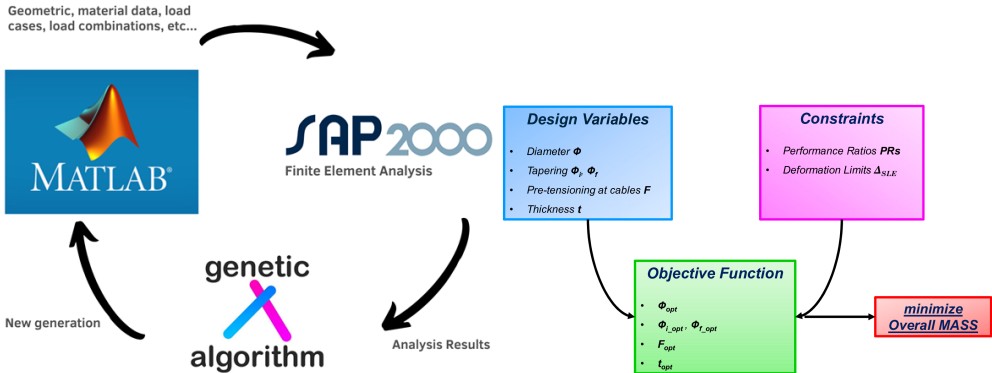

**Figure 9.** Workflow of the optimization problem.

**Table 5.** Parametric study on the design variables involved and summary of the different scenarios.

| Scenario | No. Parameters |
|---|---|
| A($\Phi$) | 1 |
| B($\Phi_i, \Phi_f$) | 2 |
| C($\Phi_i, \Phi_f, F$) | 3 |
| D($\Phi_i, \Phi_f, t$) | 3 |
| E($\Phi_i, \Phi_f, t, F$) | 4 |
| F($\Phi_i, \Phi_f, t_{ends}, t_{inter}, F$) | 5 |
| G($t_1, t_2, t_3, t_4, t_5$) | 5 |
| H($\Phi_i, \Phi_f, t_1, t_2, t_3, t_4, t_5$) | 8 |

**Table 6.** Total mass of the main pole.

| Parameter | Measure | Value |
|-----------|---------|-------|
| $\Phi_0$ | [mm] | 168.3 |
| $t_0$ | [mm] | 12.5 |
| L | [mm] | 6000 |
| Mass | [kg] | 288 |
| no elements | [-] | 5 |
| Total Mass | [kg] | 1440 |

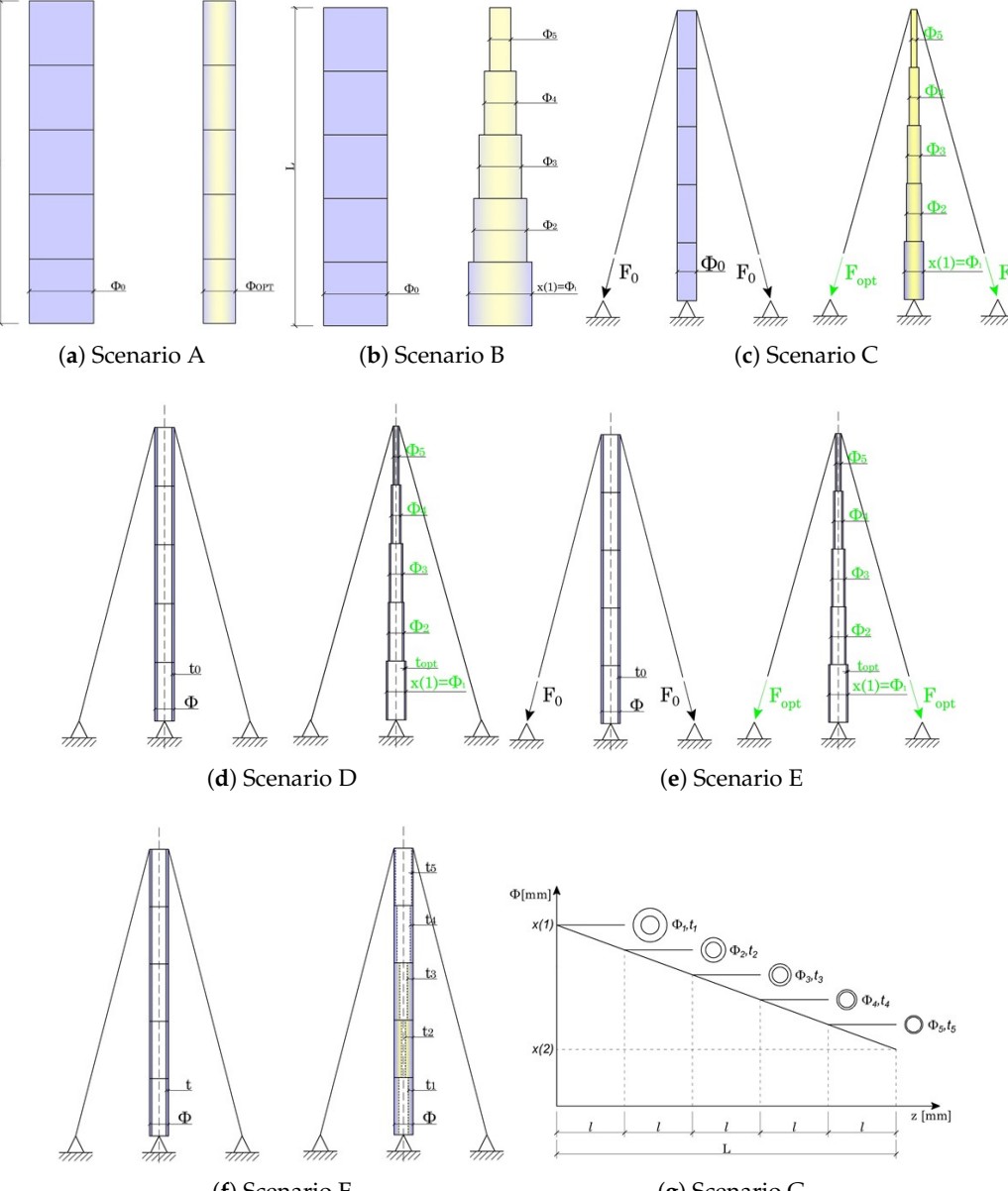

(**a**) Scenario A    (**b**) Scenario B    (**c**) Scenario C

(**d**) Scenario D    (**e**) Scenario E

(**f**) Scenario F    (**g**) Scenario G

**Figure 10.** Parametric study on the design variables involved and representation of the different scenarios described in Table 5.

*Constraints Involved in the Structural Optimization Problem*

The optimization problem statement is reported in (4) and the constraints were treated with the penalty-based approach illustrated in (5), by converting the constrained problem into an equivalent unconstrained one. The resolution of the optimization task considers the structural design assessment required by national and international codes to ensure the safety of constructions. In particular, the structural verifications derive from Eurocode 3

(EN 1993-1-1: 2005) and are referred to the ultimate limit state (ULS). The design verifications include tensile, compression, and buckling verification, and a combined assessment, such as the interaction capacity according to Annex B of the Eurocode 3:

$$\frac{D}{C} = \frac{N_{Ed}}{\frac{\chi_y A f_{yk}}{\gamma_{M1}}} + k_{yy} \frac{M_{y,Ed}}{\frac{\chi_{LT} W_{pl,y} f_{yk}}{\gamma_{M1}}} + k_{yz} \frac{M_{z,Ed}}{\frac{W_{pl,z} f_{yk}}{\gamma_{M1}}} \leq 1 \tag{8}$$

$$\frac{D}{C} = \frac{N_{Ed}}{\frac{\chi_z A f_{yk}}{\gamma_{M1}}} + k_{zy} \frac{M_{y,Ed}}{\frac{\chi_{LT} W_{pl,y} f_{yk}}{\gamma_{M1}}} + k_{zz} \frac{M_{z,Ed}}{\frac{W_{pl,z} f_{yk}}{\gamma_{M1}}} \leq 1 \tag{9}$$

where $D$ stands for the demand and $C$ stands for the capacity of the structure. Specifically, $N_{Ed}$ is the acting axial force, whereas $M_{y,Ed}$ and $M_{z,Ed}$ represent the acting bending moments in the two principal directions of a planar local reference system centered on the cross section center of gravity. $A$ is the cross section area of the pole, $W_{pl,y}$ and $W_{pl,z}$ are the plastic section modulus in the two principal directions, $f_{yk}$ is the yielding strength of the steel, whereas $\gamma_{M1}$ is the partial safety factor for instability conditions, equal to 1.05 from the Italian National Annex. $\chi_{LT}$ is the reduction factor for lateral–torsional buckling, whereas $kyy$, $kyz$, $kzy$, and $kzz$ are interaction factors whose values are derived according to two alternative approaches based on Annex A (method 1)and Annex B (method 2). The global structural deformation referred to the service limit state (SLS) has also been considered by verifying the top displacement of the mast. Specific recommendations for guyed mast structures are missing in national and international codes. Therefore, the authors adopted the suggestions defined in the Italian Technical Code NTC2018 (D.M.17/01/2018) reported in Chapter 4.2.4.2.2 Table 4.2.XIII related to limitations of lateral displacements of steel multi-storey frame structures. These limitations express a threshold condition in terms of the total height of the structure $H$:

$$\delta_{SLS,top} \leq \delta_{SLS,top,lim} = \frac{H}{500} = \frac{30000 \text{ mm}}{500} = 60 \text{ mm} \tag{10}$$

Since this condition is specific for steel multi-storey frame structures, the authors will assume this value as a reasonable choice to ensure service life assessment and preservation of working conditions of the telecommunication guyed mast tower. In the next section, a discussion on the results is carried out.

## 6. Results and Discussion

The paper compares the outcomes of the size and shape optimization in eight different scenarios, distinguished by different design variables. Scenario A is associated with the worst improvement of the structural performance since a single diameter is used for the central pole. Additionally, industrial steel profiles do not cover all possible ranges of the diameter. Improvements in the structural performance and weight reduction are achieved in the following scenarios when the search space becomes larger by increasing the number of design variables.

Scenario B introduces the tapering of the central pole with a linear variation from the bottom to the top. In this case, the optimal solution is affected by intermediate sections, which are more stressed. Consequently, the end cross-sections are over-estimated. In response to that, Scenario F introduces the linear tapering of the tube thickness $t_{ends}, t_{inter}$ to enhance the performance of the optimal solution. Parallelly, in Scenario G, five different thicknesses are adopted ($t_1, t_2, t_3, t_4, t_5$), and the results are analogue to case F. Therefore, the thickness of the steel members is a suitable optimization parameter. At the same time, the diameter alone is not capable of returning attractive solutions because a linear interpolation trend is used. In addition, lower and upper limits were imposed for $d$ and $t$. In particular, for this kind of structure, a minimum diameter $d_{min} \geq 100$ mm and a minimum thickness $t_{min} \geq 3$ mm was imposed.

The cross-section area depends on the square of the thickness. Therefore, small changes in $t$ significantly affect the resulting area. Conversely, if the diameter is the sole search space, despite being tapered linearly with height, even significant modifications may not produce notable improvements. Still, the increment of design variables involved in the structural optimization typically increases the computational efforts. However, the scenario with the highest number of variables was characterized by an average time iteration close to 18s, using a computer with average performance. The computational effort cost of the optimization procedure strongly depends on the machine performance, no convergence issues occur. Table 7 lists the average values of performance ratio obtained from the eight optimization scenarios. All scenarios were collected in terms of number of parameters involved during the analysis. Table 7 proves that the increment in the number of design variables is associated with higher performance ratios. The target of the optimization achieves the best weight reduction, fully exploiting the structural material, without exceeding the ultimate and service limit states. Table 7 lists three sets of performance ratios: the initial one before optimization, the optimized, and the one obtained using commercial steel profiles, called the design performance ratio. The averaged performance ratio is equal to 28% before optimization. It significantly increases from scenario A, nearly 45%, to scenario G with 68%.

**Table 7.** Averaged performance ratios obtained in each optimization scenario.

| No Parameters | PR Initial | PR Optimized | PR Design |
|:---:|:---:|:---:|:---:|
| | [%] | [%] | [%] |
| 1 | | 45.7 | 40.5 |
| 2 | | 39.5 | 43.1 |
| 3 | | 50.5 | 50.6 |
| 4 | 28.0 | 54.4 | 58 |
| 5 | | 65.8 | 60.2 |
| 8 | | 68 | 66 |

Essentially related to PR, mass reduction gives an idea about how much lighter (or heavier) the structure becomes due to the optimization process. It directly provides an estimate of cost savings.

Therefore, the results in Table 8 are consistent with the ones in terms of performance ratios, shown in Table 7.

**Table 8.** Mass values before/after optimization and after proper approximation (design) using commercial steel profiles.

| No Parameters | Initial Mass [kg] | Optimized Mass [kg] | Design Mass [kg] |
|:---:|:---:|:---:|:---:|
| 1 | | 1003 | 1176 |
| 2 | | 1051 | 1111 |
| 3 | | 803 | 818 |
| 4 | 1440 | 574 | 588 |
| 5 | | 403 | 453 |
| 8 | | 385 | 408 |

Figure 11 shows the optimization results for the Scenario G, in term of the performance ratio obtained by averaging the performance ratios for each structural element. The results for all scenarios are reported in Appendix A. Scenario G, depicted below, exhibits higher values of the performance ratios. This fact becomes become more evident for poles 2, 3, and 4. In these cases, the performance ratios, associated with the design solutions, achieved values equal or greater than the optimized one due to the approximation of the design section adopted. In the post-processing phase, in fact, the optimized section chosen by the list of the FE software was manually edited since the structural constraint violation or the maximum performance ratio was not reached during to the optimization process.

Moreover, in Table 9, the optimized design section for different independent iterations and the proposed industrial solutions according to product list, provided by the software, are listed. As expected, the mass reduction achieved during the optimization process results higher than the design solution due to the approximation issue. For the proposed scenario, the iteration ($N_{trial}$) that guarantees the best objective function is the second. In Appendix A, the graphical (through histogram charts) and numerical representation (through tables) of the optimization result for each scenario are provided. In order to provide an overview of the objective function trend, the performance ratios and mass reduction for each scenario were collected into Figures 12 and 13. The mentioned values were obtained for each scenario, making an average of the results, before and after optimization, independently, for each steel profile composing the central pole.

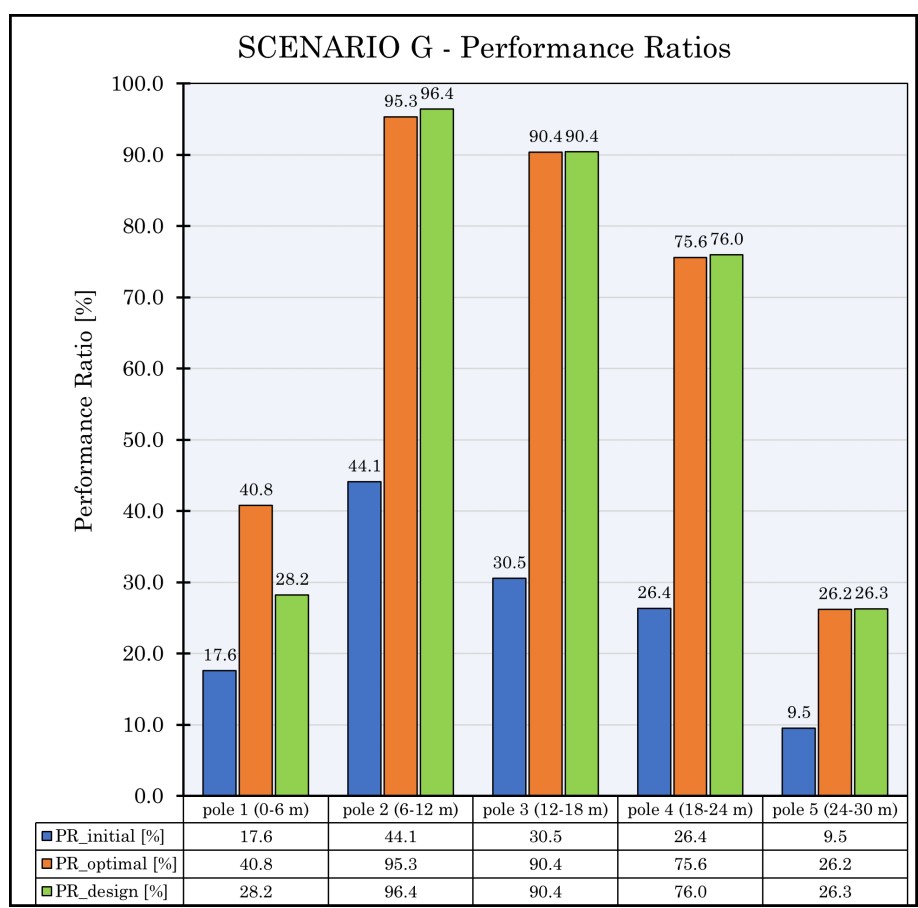

**Figure 11.** Scenario G.—PRs trend. In blue—the performance ratios of each pole before optimization are illustrated, otherwise orange for the optimized solution. In green—PRs at a design configuration according to the product list.

Figure 12 highlights an almost monotone increment of the performance ratios to the number of design variables. Interestingly, for a number of variables n ≥ 5, no significant improvements are achieved. Figure 13 emphasizes an important reduction of structural mass as the design variables increase. Once again, n = 5 represents trade-off. If the number of variables exceed 5, no significant improvements are observed.

Figures 12 and 13 show a comparison between each scenario in terms of the average performance ratio and mass reduction, respectively. Figure 12 highlights the difference with the initial state, which has an average performance ratio $PR_0 = 25.6\%$. An evident improvement is achieved for scenarios that include the thickness t as the design variable.

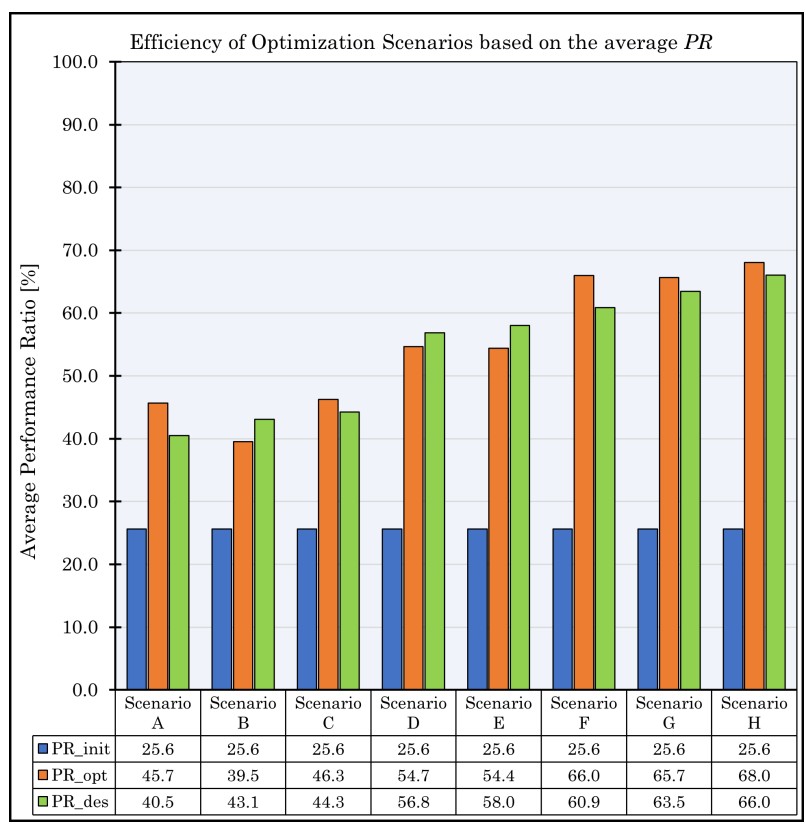

**Figure 12.** In blue, orange, and green, the average PRs, respectively, at the initial condition, after optimization, and design solution.

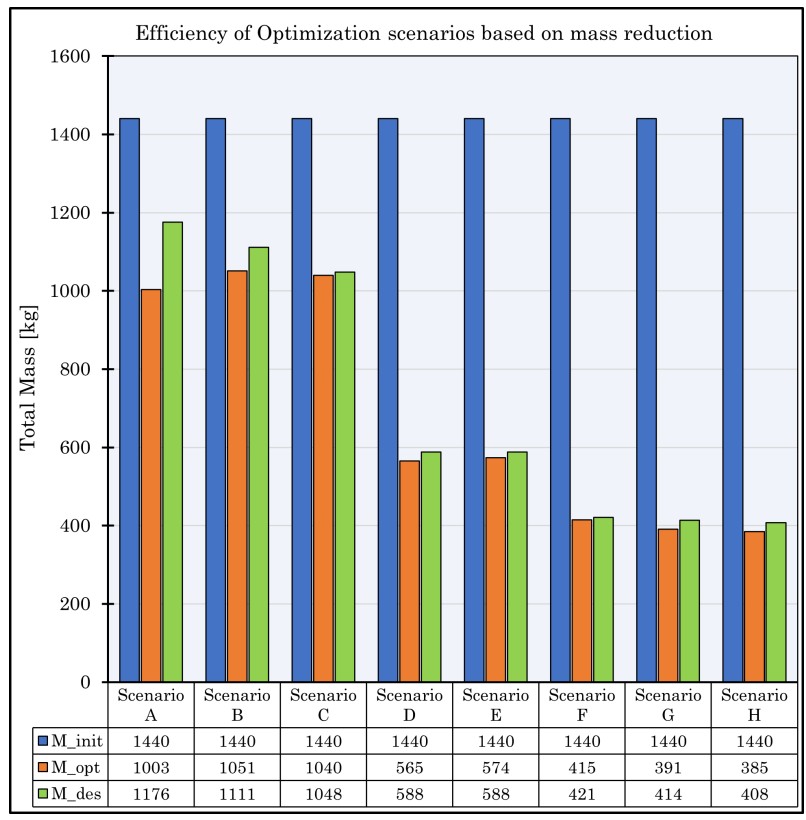

**Figure 13.** Increasing the number of design variables, the final mass becomes gradually smaller, until 385 kg (scenario H).

**Table 9.** Scenario G results: optimized solutions for the different independent executions (Ntrial) and proposed industrial one, according to the product list.

| SCENARIO G—Optimized Solution | | | | |
|---|---|---|---|---|
| Element | d [mm] | t [mm] | L [mm] | Mass [Kg] |
| Pole 1 (0–6 m) | 168.3 | 3 | 6000 | 73 |
| Pole 2 (6–12 m) | 168.3 | 4 | 6000 | 97 |
| Pole 3 (12–18 m) | 168.3 | 3 | 6000 | 73 |
| Pole 4 (18–24 m) | 168.3 | 3 | 6000 | 73 |
| Pole 5 (24–30 m) | 168.3 | 3 | 6000 | 73 |
| | Total Mass [kg] | | $\Sigma$ | 391 |
| Mass variation [kg] | −1050 | Mass variation [%] | | −72.88 |
| SCENARIO G—Design proposed according to the product list | | | | |
| Element | d [mm] | t [mm] | L [mm] | Mass [Kg] |
| Pole 1 (0–6 m) | 168.3 | 4 | 6000 | 97 |
| Pole 2 (6–12 m) | 168.3 | 4 | 6000 | 97 |
| Pole 3 (12–18 m) | 168.3 | 3 | 6000 | 73 |
| Pole 4 (18–24 m) | 168.3 | 3 | 6000 | 73 |
| Pole 5 (24–30 m) | 168.3 | 3 | 6000 | 73 |
| | Total Mass [kg] | | $\Sigma$ | 414 |
| Mass variation [kg] | −1026 | Mass variation [%] | | −71.22 |
| Ntrial = 3 | | | | | |
| $t_1$ [mm] | $t_2$ [mm] | $t_3$ [mm] | $t_4$ [mm] | $t_5$ [mm] | OF [kN] |
| 3 | 4 | 4 | 3 | 3 | 34.985 |
| 3 | 4 | 3 | 3 | 3 | 34.751 |
| 3 | 4 | 3 | 4 | 3 | 34.985 |

In particular, from Scenarios D, E, F, G, H, the average performance ratios exceed 50%, resulting in a more than 40% difference compared to the initial state. Figure 12 shows that the commercial profiles are sufficient to accommodate the optimized solution. An exception is noticeable in Scenario A because the optimization is performed using just one diameter $\Phi$, which is optimal for a few parts of the structure, while others are "over-fitted", resulting in a decrease of the performance ratios −28.4% and an increase of structural mass (+173 kg), as shown in Figure 13.

Similarly, a monotonic increment of the structural mass at the end of the optimization process is evident from Figure 13. In this case, the tonnage decreases with the increasing of the parameter's number. There is an overall mass reduction of about −67.5% (−972 kg) from scenario D to H. In scenarios A, B, and C, the thickness t of structural members is not considered. Therefore, the mass loss is not satisfactory, at about −28.4% (−409 kg). The choice of the best scenario should depend on one of the five situations described above (from D to H) related to the better PRs gain and mass loss.

## 7. Conclusions

In this paper, a guyed radio mast's size and shape optimization process was carried out to identify the equilibrium solution that guarantees the lighter optimized model, verifying strength, instability, and deformation requirements. The paper considers a detailed evaluation of the variable loads according to the Eurocodes recommendations. Furthermore, the OAPI was used to perform a structural analysis with the finite element software SAP2000 by considering the non-linearity of the cables. The optimization was

carried out using a genetic optimization algorithm. Eight scenarios (labeled from A to H) were investigated, considering different arrangements of the geometric characteristics of the central pole and cables. The input parameters were increased from Scenario A to H to achieve the best fitness value of the self-weight. From Scenario A to H, the mass reduction index generally increased with the computational effort except in scenarios B and E, in which the input parameter did not represent the best vector design for the structural optimization. At this stage, the best design solution was evaluated from the database of cross-sections inside the finite element software. Though Scenario A provides the worst structural solution in terms of objective function, it represents the most convenient optimization strategy due to its low computational effort; on the contrary, Scenario H exhibits the best fitness value with the lowest self-weight, but it represents the most time-consuming solution. The best solution is achieved when the thickness values of each member, which, composed of the central pole, are included in the optimization process. An improvement of the structural behaviour against instability is observed with increasing thickness. This verification is critical for this structure, mainly subjected to normal stresses resulting from self-weight and pre-stressing cable force. The entire optimization process seems to not be sensible to the pole diameter, chosen as the input parameter of the design vector. Although the final results of the FEM analyses are based on the Italian standards, other codes (e.g., Eurocodes, American code, etc.) can be selected from the SAP2000 settings. However, since no detailed analysis was carried out and many standards are based on the semi-probabilistic approach, the final results should be similar, even with different code formulations. Nevertheless, the partial safety factors involved in load combinations remain quite the same from the numerical point of view, regardless of the followed code.

In future developments, the authors will attempt to replace circular hollow sections with built-up steel solutions to achieve the best structural performance and assemblage procedures. Especially for higher structures, guyed radio masts generally consist of a truss skeleton. Another possible development could be a structural optimization for a cable-stayed radio antenna adopting other optimization strategies, such as particle swarm optimization, PSO, and the evolution differential algorithm (EDA), which could be less time-consuming. Finally, it could perform a typological optimization by managing the position of the cable connection, trying to find the best attachment points.

**Supplementary Materials:** The following are available online at https://www.mdpi.com/article/10.3390/app12104875/s1.

**Author Contributions:** Conceptualization, R.C., M.M.R. and J.M.; methodology, R.C., M.M.R., A.A., J.M., M.L.G. and G.C.M.; software, M.L.G., M.M.R. and R.C.; validation, R.C., M.L.G., J.M. and A.A.; formal analysis, R.C., J.M. and A.A.; investigation, A.A. and R.C.; resources, R.C., M.M.R. and G.C.M.; data curation, R.C. and A.A.; writing—original draft preparation, A.A., R.C. and M.M.R.; writing—review and editing, J.M., R.C., M.L.G. and A.A.; visualization, M.M.R., R.C. and A.A.; supervision, G.C.M. All authors have read and agreed to the published version of the manuscript.

**Funding:** This research was supported by project MSCA-RISE-2020 Marie Skłodowska-Curie Research and Innovation Staff Exchange (RISE)—ADDOPTML (ntua.gr).

**Institutional Review Board Statement:** Not applicable.

**Informed Consent Statement:** Not applicable.

**Data Availability Statement:** The data used to support the findings of this study are available from the corresponding author upon reasonable request.

**Acknowledgments:** The authors would like to thank the anonymous reviewers for their valuable comments and suggestions in revising the paper. The authors would like to thank G.C. Marano and the project ADDOPTML for funding/supporting this research.

**Conflicts of Interest:** The authors declare no conflict of interest.

## Appendix A

**Table A1.** Drag and lift forces according to [36] at ULS in [Kg/m].

| | Wind Action (Drag D, Lift L) at ULS | | | | | |
|---|---|---|---|---|---|---|
| z (m) | Drag_1 | Lift_1 | Drag_2 | Lift_2 | Drag_2 | Lift_3 |
| 1 | 3.92 | 6.5 | 6.01 | 5.2 | 6.85 | 1.95 |
| 2 | 3.92 | 6.5 | 6.01 | 5.2 | 6.85 | 1.95 |
| 3 | 3.92 | 6.5 | 6.01 | 5.2 | 6.85 | 1.95 |
| 4 | 3.92 | 6.5 | 6.01 | 5.2 | 6.85 | 1.95 |
| 5 | 3.92 | 6.5 | 6.01 | 5.2 | 6.85 | 1.95 |
| 6 | 4.2 | 6.97 | 6.44 | 5.58 | 7.34 | 2.09 |
| 7 | 4.44 | 7.37 | 6.81 | 5.9 | 7.76 | 2.21 |
| 8 | 4.66 | 7.73 | 7.15 | 6.19 | 8.14 | 2.32 |
| 9 | 4.85 | 8.05 | 7.44 | 6.44 | 8.48 | 2.42 |
| 10 | 5.03 | 8.35 | 7.71 | 6.68 | 8.79 | 2.5 |
| 11 | 5.19 | 8.61 | 7.96 | 6.89 | 9.07 | 2.58 |
| 12 | 5.34 | 8.86 | 8.19 | 7.09 | 9.33 | 2.66 |
| 13 | 5.48 | 9.09 | 8.4 | 7.27 | 9.57 | 2.73 |
| 14 | 5.61 | 9.31 | 8.6 | 7.45 | 9.8 | 2.79 |
| 15 | 5.73 | 9.51 | 8.79 | 7.61 | 10.01 | 2.85 |
| 16 | 5.85 | 9.7 | 8.97 | 7.76 | 10.22 | 2.91 |
| 17 | 5.96 | 9.88 | 9.13 | 7.91 | 10.41 | 2.97 |
| 18 | 6.06 | 10.06 | 9.29 | 8.04 | 10.59 | 3.02 |
| 19 | 6.16 | 10.22 | 9.45 | 8.18 | 10.76 | 3.07 |
| 20 | 6.25 | 10.38 | 9.59 | 8.3 | 10.92 | 3.11 |
| 21 | 6.34 | 10.53 | 9.73 | 8.42 | 11.08 | 3.16 |
| 22 | 6.43 | 10.67 | 9.86 | 8.54 | 11.23 | 3.2 |
| 23 | 6.51 | 10.81 | 9.99 | 8.65 | 11.38 | 3.24 |
| 24 | 6.59 | 10.94 | 10.11 | 8.75 | 11.52 | 3.28 |
| 25 | 6.67 | 11.07 | 10.23 | 8.86 | 11.66 | 3.32 |
| 26 | 6.75 | 11.19 | 10.35 | 8.96 | 11.79 | 3.36 |
| 27 | 6.82 | 11.31 | 10.46 | 9.05 | 11.91 | 3.39 |
| 28 | 6.89 | 11.43 | 10.56 | 9.14 | 12.03 | 3.43 |
| 29 | 6.96 | 11.54 | 10.67 | 9.23 | 12.15 | 3.46 |
| 30 | 7.02 | 11.65 | 10.77 | 9.32 | 12.27 | 3.5 |

**Table A2.** Drag and lift forces according to [36] at SLS in [Kg/m].

| | Wind Action (Drag D, Lift L) at SLS | | | | | |
|---|---|---|---|---|---|---|
| z (m) | Drag_1 | Lift_1 | Drag_2 | Lift_2 | Drag_2 | Lift_3 |
| 1 | 2.29 | 3.81 | 3.52 | 3.05 | 4.01 | 1.14 |
| 2 | 2.29 | 3.81 | 3.52 | 3.05 | 4.01 | 1.14 |
| 3 | 2.29 | 3.81 | 3.52 | 3.05 | 4.01 | 1.14 |
| 4 | 2.29 | 3.81 | 3.52 | 3.05 | 4.01 | 1.14 |
| 5 | 2.29 | 3.81 | 3.52 | 3.05 | 4.01 | 1.14 |
| 6 | 2.31 | 3.84 | 3.54 | 3.07 | 4.04 | 1.15 |
| 7 | 2.32 | 3.86 | 3.57 | 3.09 | 4.06 | 1.16 |
| 8 | 2.34 | 3.88 | 3.58 | 3.1 | 4.08 | 1.16 |
| 9 | 2.34 | 3.89 | 3.6 | 3.11 | 4.1 | 1.17 |

**Table A2.** *Cont.*

| | | Wind Action (Drag D, Lift L) at SLS | | | |
|---|---|---|---|---|---|
| z (m) | Drag_1 | Lift_1 | Drag_2 | Lift_2 | Drag_2 | Lift_3 |
| 10 | 2.35 | 3.9 | 3.61 | 3.12 | 4.11 | 1.17 |
| 11 | 2.36 | 3.92 | 3.62 | 3.13 | 4.12 | 1.17 |
| 12 | 2.37 | 3.93 | 3.63 | 3.14 | 4.13 | 1.18 |
| 13 | 2.37 | 3.94 | 3.64 | 3.15 | 4.14 | 1.18 |
| 14 | 2.38 | 3.94 | 3.64 | 3.15 | 4.15 | 1.18 |
| 15 | 2.38 | 3.95 | 3.65 | 3.16 | 4.16 | 1.19 |
| 16 | 2.39 | 3.96 | 3.66 | 3.17 | 4.17 | 1.19 |
| 17 | 2.39 | 3.96 | 3.66 | 3.17 | 4.17 | 1.19 |
| 18 | 2.39 | 3.97 | 3.67 | 3.18 | 4.18 | 1.19 |
| 19 | 2.4 | 3.98 | 3.67 | 3.18 | 4.19 | 1.19 |
| 20 | 2.4 | 3.98 | 3.68 | 3.19 | 4.19 | 1.19 |
| 21 | 2.4 | 3.99 | 3.68 | 3.19 | 4.2 | 1.2 |
| 22 | 2.4 | 3.99 | 3.69 | 3.19 | 4.2 | 1.2 |
| 23 | 2.41 | 4 | 3.69 | 3.2 | 4.21 | 1.2 |
| 24 | 2.41 | 4 | 3.7 | 3.2 | 4.21 | 1.2 |
| 25 | 2.41 | 4 | 3.7 | 3.2 | 4.21 | 1.2 |
| 26 | 2.41 | 4.01 | 3.7 | 3.21 | 4.22 | 1.2 |
| 27 | 2.42 | 4.01 | 3.71 | 3.21 | 4.22 | 1.2 |
| 28 | 2.42 | 4.01 | 3.71 | 3.21 | 4.23 | 1.2 |
| 29 | 2.42 | 4.02 | 3.71 | 3.21 | 4.23 | 1.21 |
| 30 | 2.42 | 4.02 | 3.72 | 3.22 | 4.23 | 1.21 |

**Table A3.** Modal participating mass ratios.

| | Modal Participating Mass Ratios | | | |
|---|---|---|---|---|
| n. Modes | Period (s) | Frequence (Hz) | Part. Mass X (%) | Part. Mass Y [%] |
| 1 | 3.99 | 0.251 | 0.0 | 0.28 |
| 2 | 3.99 | 0.251 | 0.83 | 0.0 |
| 3 | 3.99 | 0.251 | 0.0 | 0.55 |
| 4 | 3.473 | 0.288 | 0.0 | 0.25 |
| 5 | 3.473 | 0.288 | 0.75 | 0.0 |
| 6 | 3.472 | 0.288 | 0.0 | 0.5 |
| 7 | 2.929 | 0.341 | 0.0 | 2.52 |
| 8 | 2.925 | 0.342 | 0.0 | 0.05 |
| 9 | 2.916 | 0.343 | 2.55 | 0.0 |
| 10 | 0.437 | 2.290 | 9.58 | 26.24 |
| 11 | 0.434 | 2.304 | 26.41 | 9.16 |
| 12 | 0.206 | 4.853 | 7.16 | 4.44 |
| 13 | 0.203 | 4.934 | 5.64 | 4.78 |
| 14 | 0.155 | 6.437 | 27.38 | 0.78 |
| 15 | 0.144 | 6.935 | 3.67 | 26.10 |
| 16 | 0.116 | 8.584 | 0.14 | 0.00 |
| 17 | 0.116 | 8.595 | 0.00 | 0.32 |
| 18 | 0.106 | 9.436 | 2.35 | 9.82 |
| 19 | 0.057 | 17.410 | 0.03 | 0.08 |
| 20 | 0.057 | 17.442 | 0.01 | 0.01 |
| 21 | 0.054 | 18.573 | 0.48 | 0.45 |
| 22 | 0.050 | 20.163 | 0.03 | 0.00 |
| 23 | 0.047 | 21.496 | 0.00 | 0.00 |

**Table A3.** *Cont.*

| | | Modal Participating Mass Ratios | | |
|---|---|---|---|---|
| **n. Modes** | **Period (s)** | **Frequence (Hz)** | **Part. Mass X (%)** | **Part. Mass Y (%)** |
| 24 | 0.046 | 21.516 | 0.00 | 0.00 |
| 25 | 0.036 | 27.906 | 0.27 | 0.11 |
| 26 | 0.035 | 28.313 | 0.04 | 0.13 |
| 27 | 0.032 | 31.224 | 0.01 | 0.48 |
| 28 | 0.032 | 31.722 | 0.20 | 0.00 |
| 29 | 0.031 | 32.590 | 0.01 | 0.06 |
| 30 | 0.024 | 40.831 | 0.00 | 0.00 |
| 31 | 0.024 | 40.836 | 0.00 | 0.00 |
| 32 | 0.023 | 42.600 | 12.32 | 0.00 |
| 33 | 0.022 | 44.518 | 0.01 | 10.59 |

**Table A4.** Load combination.

| | Load Combination |
|---|---|
| ULS Max$_1$ | $1.3 \cdot G_1 + 1.5 \cdot G_2 + 1.5 \cdot Wind_1 + 1.5 \cdot 0.5 \cdot Ice_1 + 1.5 \cdot 0 \cdot Q_M$ |
| ULS Max1$_2$ | $1.3 \cdot G_1 + 1.5 \cdot G_2 + 1.5 \cdot Q_M + 1.5 \cdot 0.6 Wind_1 + 1.5 \cdot 0.2 Ice_1$ |
| ULS Min$_1$ | $1 \cdot G_1 + 0.8 \cdot G_2 + 1.5 \cdot Wind_1 + 1.5 \cdot 0.5 Ice_1 + 1.5 \cdot 0 Q_M$ |
| Quake$_1$ | $E + G_1 + G_2 + 0 \cdot Wind_1 + 0 Ice_1 + 0 Q_M$ |
| ULS Max$_2$ | $1.3 \cdot G_1 + 1.5 \cdot G_2 + 1.5 \cdot Wind_2 + 1.5 \cdot 0.5 \cdot Ice_2 + 1.5 \cdot 0 \cdot Q_M$ |
| ULS Max2$_1$ | $1.3 \cdot G_1 + 1.5 \cdot G_2 + 1.5 \cdot Q_M + 1.5 \cdot 0.6 Wind_2 + 1.5 \cdot 0.2 Ice_2$ |
| ULS Min$_1$ | $1 \cdot G_1 + 0.8 \cdot G_2 + 1.5 \cdot Wind_2 + 1.5 \cdot 0.5 Ice_2 + 1.5 \cdot 0 Q_M$ |
| Quake$_2$ | $E + G_1 + G_2 + 0.8 \cdot Wind_2 + 1.5 Ice_2 + 1.5 Q_M$ |

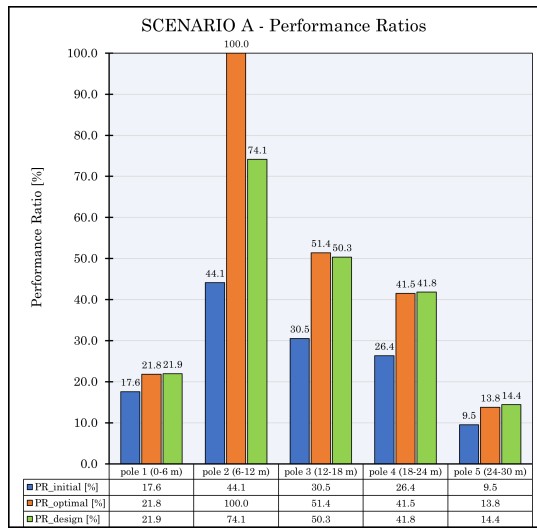

**Figure A1.** Cont.

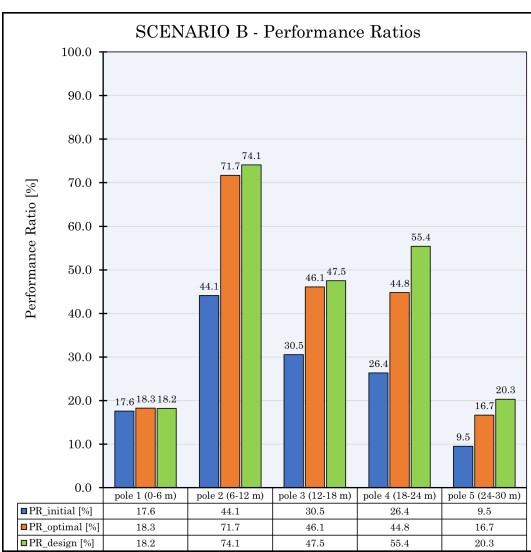

**Figure A1.** Scenarios A, B. In blue, orange, and green, the average PRs, respectively, at the initial condition, after optimization, and the design solution.

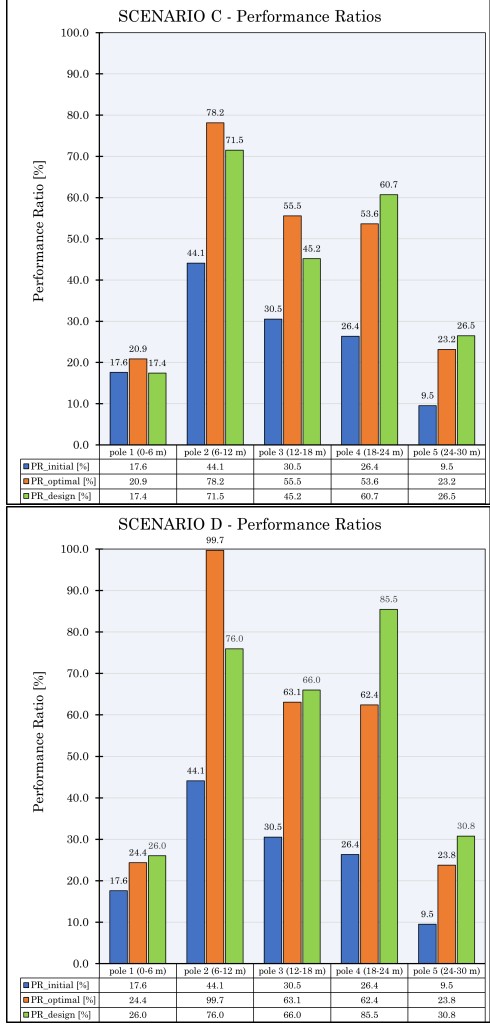

**Figure A2.** Scenarios C, D. In blue, orange, and green, the average PRs, respectively, at the initial condition, after optimization, and the design solution.

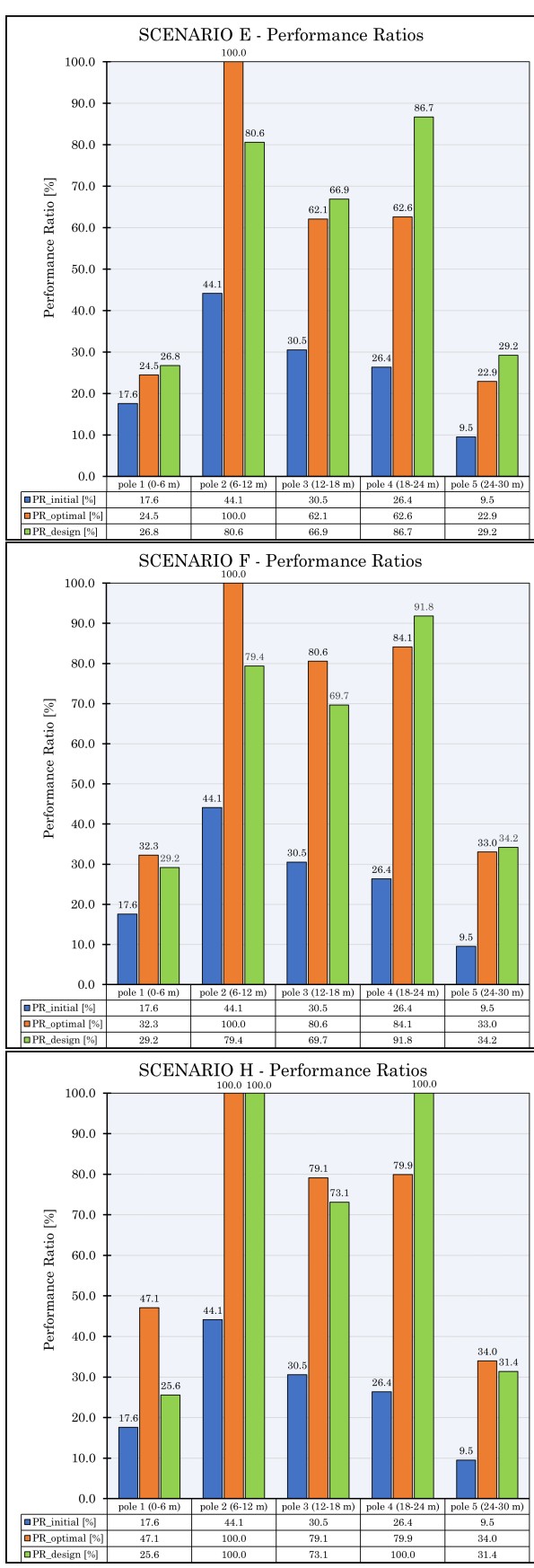

**Figure A3.** Scenarios E, F, H. In blue, orange and green, the average PRs, respectively, at the initial condition, after optimization, and the design solution.

**Table A5.** Scenario A results: optimized solutions for the different independent executions (Ntrial) and the proposed industrial one, according to the product list.

| SCENARIO A—Optimized Solution | | | | |
|---|---|---|---|---|
| Element | d [mm] | t [mm] | L [mm] | Mass [Kg] |
| Pole 1 (0–6 m) | 121 | 12.5 | 6000 | 201 |
| Pole 2 (6–12 m) | 121 | 12.5 | 6000 | 201 |
| Pole 3 (12–18 m) | 121 | 12.5 | 6000 | 201 |
| Pole 4 (18–24 m) | 121 | 12.5 | 6000 | 201 |
| Pole 5 (24–30 m) | 121 | 12.5 | 6000 | 201 |
| | | Total Mass [kg] | Σ | 1003 |
| Mass variation [kg] | −437 | Mass variation [%] | | −30.36 |
| **SCENARIO A—Design proposed according to the product list** | | | | |
| Element | d [mm] | t [mm] | L [mm] | Mass [Kg] |
| Pole 1 (0–6 m) | 139.7 | 12.5 | 6000 | 235 |
| Pole 2 (6–12 m) | 139.7 | 12.5 | 6000 | 235 |
| Pole 3 (12–18 m) | 139.7 | 12.5 | 6000 | 235 |
| Pole 4 (18–24 m) | 139.7 | 12.5 | 6000 | 235 |
| Pole 5 (24–30 m) | 139.7 | 12.5 | 6000 | 235 |
| | | Total Mass [kg] | Σ | 1176 |
| Mass variation [kg] | −264 | Mass variation [%] | | −18.36 |
| **Ntrial = 5** | | | | |
| $\Phi_{opt}$ [mm] | | | OF [kN] | |
| **121** | | | 40.758 | |
| 121 | | | 40.758 | |
| 121 | | | 40.758 | |
| 121 | | | **40.758** | |
| 122 | | | 40.849 | |

**Table A6.** Scenario B results: optimized solutions for the different independent executions (Ntrial) and the proposed industrial one according to the product list.

| SCENARIO B—Optimized Solution | | | | |
|---|---|---|---|---|
| Element | d [mm] | t [mm] | L [mm] | Mass [Kg] |
| Pole 1 (0–6 m) | 149 | 12.5 | 6000 | 252 |
| Pole 2 (6–12 m) | 138 | 12.5 | 6000 | 231 |
| Pole 3 (12–18 m) | 126 | 12.5 | 6000 | 210 |
| Pole 4 (18–24 m) | 115 | 12.5 | 6000 | 189 |
| Pole 5 (24–30 m) | 103 | 12.5 | 6000 | 168 |
| | | Total Mass [kg] | Σ | 1051 |
| Mass variation [kg] | −389 | Mass variation [%] | | −27.02 |
| **SCENARIO B—Design proposed according to product list** | | | | |
| Element | d [mm] | t [mm] | L [mm] | Mass [Kg] |
| Pole 1 (0–6 m) | 168.3 | 12.5 | 6000 | 288 |
| Pole 2 (6–12 m) | 139.7 | 12.5 | 6000 | 235 |
| Pole 3 (12–18 m) | 139.7 | 12.5 | 6000 | 235 |
| Pole 4 (18–24 m) | 114.3 | 12.5 | 6000 | 188 |

**Table A6.** *Cont.*

| Pole 5 (24–30 m) | 101.6 | 12.5 | 6000 | 165 |
|---|---|---|---|---|
| | Total Mass [kg] | | Σ | 1111 |
| Mass variation [kg] | −329 | Mass variation [%] | | −22.84 |

| **Ntrial = 5; best solutions** | | |
|---|---|---|
| $\Phi_i$ [mm] | $\Phi_f$ [mm] | OF [kN] |
| 148 | 94 | 41.248 |
| 146 | 103 | 41.466 |
| 148 | 94 | 41.248 |
| 146 | 103 | 41.466 |
| **149** | **92** | **41.230** |

**Table A7.** Scenario C results: optimized solutions for the different independent executions (Ntrial) and the proposed industrial one according to the product list.

| **SCENARIO C—Optimized solution** | | | | |
|---|---|---|---|---|
| Element | d [mm] | t [mm] | L [mm] | Mass [Kg] |
| Pole 1 (0–6 m) | 147 | 12.5 | 6000 | 249 |
| Pole 2 (6–12 m) | 136 | 12.5 | 6000 | 228 |
| Pole 3 (12–18 m) | 125 | 12.5 | 6000 | 208 |
| Pole 4 (18–24 m) | 114 | 12.5 | 6000 | 188 |
| Pole 5 (24–30 m) | 103 | 12.5 | 6000 | 167 |
| | Total Mass [kg] | | Σ | 1040 |
| Mass variation [kg] | −400 | Mass variation [%] | | −27.79 |
| **SCENARIO C—Design proposed according to the product list** | | | | |
| Element | d [mm] | t [mm] | L [mm] | Mass [Kg] |
| Pole 1 (0–6 m) | 168.3 | 12.5 | 6000 | 288 |
| Pole 2 (6–12 m) | 139.7 | 12.5 | 6000 | 235 |
| Pole 3 (12–18 m) | 139.7 | 12.5 | 6000 | 235 |
| Pole 4 (18–24 m) | 114.3 | 10 | 6000 | 154 |
| Pole 5 (24–30 m) | 101.6 | 10 | 6000 | 135 |
| | Total Mass [kg] | | Σ | 1048 |
| Mass variation [kg] | −392 | Mass variation [%] | | −27.22 |
| **Ntrial = 5** | | | | |
| $\Phi_i$ [mm] | | $\Phi_f$ [mm] | F [kN] | OF [kN] |
| 152 | | 92 | 1.8 | 41.393 |
| 151 | | 92 | 1.4 | 41.339 |
| 149 | | 92 | 1 | 41.230 |
| 156 | | 92 | 2.4 | 41.610 |
| **147** | | **92** | **0.8** | **41.121** |

**Table A8.** Scenario D results: optimized solutions for the different independent executions (Ntrial) and the proposed industrial one according to the product list.

| SCENARIO D—Optimized solution | | | | |
|---|---|---|---|---|
| Element | d [mm] | t [mm] | L [mm] | Mass [Kg] |
| Pole 1 (0–6 m) | 161 | 6 | 6000 | 138 |
| Pole 2 (6–12 m) | 147 | 6 | 6000 | 125 |
| Pole 3 (12–18 m) | 133 | 6 | 6000 | 113 |
| Pole 4 (18–24 m) | 120 | 6 | 6000 | 101 |
| Pole 5 (24–30 m) | 106 | 6 | 6000 | 89 |
| | | Total Mass [kg] | Σ | 565 |
| Mass variation [kg] | −875 | Mass variation [%] | | −60.75 |
| SCENARIO D—Design proposed according to the product list | | | | |
| Element | d [mm] | t [mm] | L [mm] | Mass [Kg] |
| Pole 1 (0–6 m) | 168.3 | 6 | 6000 | 144 |
| Pole 2 (6–12 m) | 168.3 | 6 | 6000 | 144 |
| Pole 3 (12–18 m) | 139.7 | 6 | 6000 | 119 |
| Pole 4 (18–24 m) | 114.3 | 6 | 6000 | 96 |
| Pole 5 (24–30 m) | 101.6 | 6 | 6000 | 85 |
| | | Total Mass [kg] | Σ | 588 |
| Mass variation [kg] | −853 | Mass variation [%] | | −59.20 |
| Ntrial = 5 | | | | |
| $\Phi_i$ [mm] | | $\Phi_f$ [mm] | t [mm] | OF [kN] |
| 161 | | 92 | 6 | **36.465** |
| 146 | | 117 | 7 | 37.389 |
| 162 | | 92 | 6 | 36.491 |
| 162 | | 92 | 6 | 36.491 |
| 163 | | 92 | 6 | 36.517 |

**Table A9.** Scenario E results: optimized solutions for the different independent executions (Ntrial) and the proposed industrial one according to the product list.

| SCENARIO E—Optimized Solution | | | | |
|---|---|---|---|---|
| Element | d [mm] | t [mm] | L [mm] | Mass [Kg] |
| Pole 1 (0–6 m) | 165 | 6 | 6000 | 141 |
| Pole 2 (6–12 m) | 150 | 6 | 6000 | 128 |
| Pole 3 (12–18 m) | 135 | 6 | 6000 | 115 |
| Pole 4 (18–24 m) | 121 | 6 | 6000 | 102 |
| Pole 5 (24–30 m) | 106 | 6 | 6000 | 89 |
| | | Total Mass [kg] | Σ | 574 |
| Mass variation [kg] | −866 | Mass variation [%] | | −60.13 |
| SCENARIO E—Design proposed according to the product list | | | | |
| Element | d [mm] | t [mm] | L [mm] | Mass [Kg] |
| Pole 1 (0–6 m) | 168.3 | 6 | 6000 | 144 |
| Pole 2 (6–12 m) | 168.3 | 6 | 6000 | 144 |
| Pole 3 (12–18 m) | 139.7 | 6 | 6000 | 119 |

**Table A9.** *Cont.*

| | | | | |
|---|---|---|---|---|
| Pole 4 (18–24 m) | 114.3 | 6 | 6000 | 96 |
| Pole 5 (24–30 m) | 101.6 | 6 | 6000 | 85 |
| | Total Mass [kg] | | Σ | 588 |
| Mass variation [kg] | −853 | Mass variation [%] | | −59.20 |

| | | | | |
|---|---|---|---|---|
| **Ntrial = 5; best solutions** | | | | |
| $\Phi_i$ [mm] | $\Phi_f$ [mm] | t [mm] | F [kN] | OF [kN] |
| 150 | 97 | 7.8 | 1.3 | 37.766 |
| 153 | 112 | 6.4 | 1.6 | 36.964 |
| 165 | 91 | 6 | 2.3 | 36.552 |
| 160 | 91 | 7 | 1.3 | 37.287 |
| 139 | 104 | 8.8 | 1.3 | 38.337 |

**Table A10.** Scenario F results: optimized solutions for the different independent executions (Ntrial) and the proposed industrial one according to the product list.

| SCENARIO F—Optimized Solution | | | | |
|---|---|---|---|---|
| Element | d [mm] | t [mm] | L [mm] | Mass [Kg] |
| Pole 1 (0–6 m) | 157 | 4 | 6000 | 91 |
| Pole 2 (6–12 m) | 144 | 6 | 6000 | 122 |
| Pole 3 (12–18 m) | 131 | 4 | 6000 | 75 |
| Pole 4 (18–24 m) | 118 | 4 | 6000 | 67 |
| Pole 5 (24–30 m) | 105 | 4 | 6000 | 60 |
| | Total Mass [kg] | | Σ | 415 |
| Mass variation [kg] | −1025 | Mass variation [%] | | −71.16 |

| SCENARIO F—Design proposed according to the product list | | | | |
|---|---|---|---|---|
| Element | d [mm] | t [mm] | L [mm] | Mass [Kg] |
| Pole 1 (0–6 m) | 168.3 | 4 | 6000 | 97 |
| Pole 2 (6–12 m) | 168.3 | 5 | 6000 | 121 |
| Pole 3 (12–18 m) | 139.7 | 4 | 6000 | 80 |
| Pole 4 (18–24 m) | 114.3 | 4 | 6000 | 65 |
| Pole 5 (24–30 m) | 101.6 | 4 | 6000 | 58 |
| | Total Mass [kg] | | Σ | 421 |
| Mass variation [kg] | −1019 | Mass variation [%] | | −70.75 |

| Ntrial = 3; best solutions | | | | | |
|---|---|---|---|---|---|
| $\Phi_i$ [mm] | $\Phi_f$ [mm] | $t_{ends}$ [mm] | $t_{inter}$ [mm] | F [kN] | OF [kN] |
| 155 | 92 | 4 | 7 | 3.2 | 35.141 |
| 157 | 92 | 4 | 6 | 0.9 | 34.993 |
| 151 | 92 | 4 | 7 | 1.3 | 35.058 |

**Table A11.** Scenario H results: optimized solutions for the different independent executions (Ntrial) and the proposed industrial one according to the product list.

| SCENARIO H—Optimized Solution | | | | |
|---|---|---|---|---|
| Element | d [mm] | t [mm] | L [mm] | Mass [Kg] |
| Pole 1 (0–6 m) | 158 | 3 | 6000 | 69 |
| Pole 2 (6–12 m) | 146 | 6 | 6000 | 124 |
| Pole 3 (12–18 m) | 133 | 4 | 6000 | 76 |
| Pole 4 (18–24 m) | 121 | 4 | 6000 | 69 |
| Pole 5 (24–30 m) | 108 | 3 | 6000 | 47 |
| | | Total Mass [kg] | Σ | 385 |
| Mass variation [kg] | −1055 | Mass variation [%] | | −73.27 |

| SCENARIO H—Design proposed according to the product list | | | | |
|---|---|---|---|---|
| Element | d [mm] | t [mm] | L [mm] | Mass [Kg] |
| Pole 1 (0–6 m) | 168.3 | 5 | 6000 | 121 |
| Pole 2 (6–12 m) | 168.3 | 4 | 6000 | 97 |
| Pole 3 (12–18 m) | 139.7 | 4 | 6000 | 80 |
| Pole 4 (18–24 m) | 139.7 | 3 | 6000 | 61 |
| Pole 5 (24–30 m) | 114.3 | 3 | 6000 | 49 |
| | | Total Mass [kg] | Σ | 408 |
| Mass variation [kg] | −1032 | Mass variation [%] | | −71.65 |

| Ntrial = 3 | | | | | | | | |
|---|---|---|---|---|---|---|---|---|
| $\Phi_i$ [mm] | $\Phi_i$ [mm] | $t_1$ [mm] | $t_2$ [mm] | $t_3$ [mm] | $t_4$ [mm] | $t_5$ [mm] | F [kN] | OF [kN] |
| 164 | 109 | 4 | 5 | 4 | 3 | 3 | 0.9 | 34.789 |
| 167 | 111 | 3 | 6 | 4 | 3 | 3 | 2 | 34.839 |
| 158 | 96 | 3 | 6 | 4 | 4 | 3 | 2 | 34.695 |

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
