# Peer review of "Size and Shape Optimization of a Guyed Mast Structure under Wind, Ice and Seismic Loading"

_applsci, doi:10.3390/app12104875_

Round 1

Reviewer 1 Report

It's interesting that this manuscript focuses on size and shape optimization of a guyed mast structure under wind, ice and seismic loading. This manuscript discusses the size and shape optimization of a guyed radio mast for radiocommunications. The considered structure represents a widely industrial solution due to the recent spreading of the 5G and 6G mobile networks. However there are some major problems with this manuscript, as following,

What is the stress-strain curve of the components that make up the structure which is a guyed radio mast?

In the numerical model, where is the plastic hinge set?

Some of the research addressing these issues should be acknowledged, some recommended references, among many others are: https://doi.org/10.1016/j.jseaes.2013.05.008, and  https://doi.org/10.1016/j.istruc.2020.12.089,

The probability of simultaneous occurrence of wind load, snow load and earthquake load is small. How to consider the reduction between them?

So, it is  my opinion that the manuscript can be accepted in the journal by providing major revisions mainly devoted to improve its quality. 

Author Response

It's interesting that this manuscript focuses on size and shape optimization of a guyed mast structure under wind, ice and seismic loading. This manuscript discusses the size and shape optimization of a guyed radio mast for radiocommunications. The considered structure represents a widely industrial solution due to the recent spreading of the 5G and 6G mobile networks. However there are some major problems with this manuscript, as following,

What is the stress-strain curve of the components that make up the structure which is a guyed radio mast?

The authors are grateful for this comment. The steel adopted is the structural steel S355. The authors provided the stress-strain curve from SAP2000 within paragraph 3.1 Dead Loads, where all the other technical information are provided. The following sentence has been revised:

“The structure is made of steel S355, whose mechanical stress-strain behaviour is depicted in Figure 3, and the characteristics are listed here […].”

In the numerical model, where is the plastic hinge set?

The authors are grateful for this comment. In this work, linear analyses have been mainly conducted, since the main goal was the investigations of the optimization framework for guyed mast structures. Linear elastic elements have been adopted and concentrated nor distributed plasticity implementations have not been considered. Therefore, there is no need to explicitly provide the plastic hinge set. In fact, geometric nonlinearities have been considered only for the cable elements, and for the seismic analysis, the Linear Dynamic Analysis (Dynamic modal analysis) has been conducted, in which any information related to the plastic hinge set of the model cannot be pointed out. Moreover, according to Italian Standard Regulation (chapter 7.2), all the requirements, either in terms of geometric ratio or natural vibration frequency, requested to perform this specific analysis are satisfied by the investigated application case. According to the Authors, the guyed mast structure remains in the elastic stage during the seismic event, therefore, currently, there is actually no need for setting the plastic hinge set. However, in future works, nonlinear analyses such as the Pushover analysis may represent an interesting further development.

Some of the research addressing these issues should be acknowledged, some recommended references, among many others are: https://doi.org/10.1016/j.jseaes.2013.05.008, and  https://doi.org/10.1016/j.istruc.2020.12.089

The authors are grateful for this comment. The recommended references are being included within the introduction section of the present work, when seismic analyses and earthquake phenomena are discussed.

“Guyed masts are extensively used in the telecommunications industry, and their size, shape and topology optimization can significantly benefit their transportation and installation. The main loads acting on guyed mast structures arise from wind [ 1, 2], earthquakes [3– 6], sudden rupture of guys [7 ], galloping of guys [ 8], sudden ice shedding from ice-laden 17 guy wires [9].”

  1. Liu, C.; Fang, D.; Zhao, L. Reflection on earthquake damage of buildings in 2015 Nepal earthquake and seismic measures for post-earthquake reconstruction. Structures 2021, 30, 647–658. doi:https://doi.org/10.1016/j.istruc.2020.12.089.
  2. Sun, X.; Tao, X.; Duan, S.; Liu, C. Kappa (k) derived from accelerograms recorded in the 2008 Wenchuan mainshock, Sichuan, China. Journal of Asian Earth Sciences 2013, 73, 306–316. doi:https://doi.org/10.1016/j.jseaes.2013.05.008.

The probability of simultaneous occurrence of wind load, snow load and earthquake load is small. How to consider the reduction between them?

The authors are grateful for this comment. According to the Italian Standard Regulation NTC2018, the most critical load combinations have been performed both at the ultimate limit state (ULS) and the life safety (LS) for seismic combination, as summarized in Table A4. Italian Standard Regulation recommends the adoption of suitable safety factors for wind and snow actions. Specifically, in each load combination, the simultaneous occurrence of horizontal and vertical actions is managed with partial safety factors  and combination coefficients . These latters actually consider the probability of occurrence of the variable action with its maximum intensity. A revised paragraph was added in section 3 Load Analysis and, in appendix A, a new table with the most critical Load Combinations adopted into the analysis is reported.

“According to the Italian Standard Regulation NTC2018, the load combinations of the actions have been evaluated at the ultimate limit state (ULS) and, for seismic conditions, at the life safety (LS) limit state. In appendix A, Table A4 illustrates the most critical combinations for both static and dynamic configurations. Partial safety factors γ and combination coefficients ψ have been adopted in order to consider maximization (positive sign) or minimization (negative sign) of effects both for vertical and horizontal actions.”

Reviewer 2 Report

This paper deals with the optimization of the shape of a tall thin vertical structure which is the guyed mast. During the optimization stage, all external factors specific to a given structure, such as wind force or ice loading, were taken into account. With the use of Matlab and SAP2000 FEA software, the optimization problem was correctly defined. The authors presented the current state of arts properly and well-motivated the purpose of the work. The undertaken topic is topical and worth investigating. The obtained results were provided with a broad chapter of discussion and further research plans were characterized. Due to the above, I believe that the work should be approved for publication in Applied Sciences after minor revision.

Minor remarks:

- There are a few linguistic errors in the article (eg "three main research mainstreams"). I suggest reading the article again before resubmission.

- The quality of some drawings should be improved (Fig. 4-6).

- I would also suggest referring to other  norms and defining in a few sentences whether the presented method will be adequate and simple (also outside Italy).

Author Response

This paper deals with the optimization of the shape of a tall thin vertical structure which is the guyed mast. During the optimization stage, all external factors specific to a given structure, such as wind force or ice loading, were taken into account. With the use of Matlab and SAP2000 FEA software, the optimization problem was correctly defined. The authors presented the current state of arts properly and well-motivated the purpose of the work. The undertaken topic is topical and worth investigating. The obtained results were provided with a broad chapter of discussion and further research plans were characterized. Due to the above, I believe that the work should be approved for publication in Applied Sciences after minor revision.

Minor remarks:

There are a few linguistic errors in the article (eg "three main research mainstreams"). I suggest reading the article again before resubmission.

The authors are grateful for this comment. The authors revised that part and checked and revised the entire manuscript.

The quality of some drawings should be improved (Fig. 4-6).

The authors are grateful for this comment. The quality and the readability of the information provided by those Figures have been improved.

I would also suggest referring to other norms and defining in a few sentences whether the presented method will be adequate and simple (also outside Italy).

The authors are grateful for this comment. In the conclusion section, some revised sentences attempt to provide a qualitative comparison among Italian Code NTC18 with other acknowledged international structural codes such as Eurocodes, American Codes. Specifically, since the semi-probabilistic approach is acknowledged as a reliable structural design method by the most popular standards, the final results from the proposed optimization framework should be quite similar even with different code formulations. The main different point among the codes could be related to the choice of the partial safety factors. For instance, starting from the partial safety factors values proposed by the Eurocode suggestions, the Italian Annex provides sometimes some more conservative values. However, numerically speaking, these values are quite the same, therefore the Italian Regulations provide very similar results with respect e.g. to the Eurocode.

“Although the final results of the FEM analyses are based on the Italian standards, other codes (e.g. Eurocodes, American code, etc.) can be selected from the SAP2000 settings. However, since no detailed analysis was carried out and many standards are based on the semi-probabilistic approach, the final results should be similar, even with different code formulations. Nevertheless, the partial safety factors involved in load combinations remain quite the same from the numerical point of view, regardless of the followed code.”

Round 2

Reviewer 1 Report

The authors have addressed most of the comments raised by the reviewers satisfactorily. Now, the manuscript is well written and the topic interesting and worth of investigation.I think it can be accepted as it is.